**Effects of passive storage conceptualization on modelling hydrological function and isotope dynamics in the flow system of cockpit karst landscape**

Guangxuan Li[1,2], Xi Chen[1,2*], Zhicai Zhang[3], Lichun Wang[1,2], Chris Soulsby[4]

**Affiliations:**

[1] Institute of Surface-Earth System Science, School of Earth System Science, Tianjin University, Tianjin, China, 300072

[2] Tianjin Key Laboratory of Earth Critical Zone Science and Sustainable Development in Bohai Rim, China

[3] College of Hydrology and Water Resources, Hohai University, Nanjing, China

[4] School of Geosciences, University of Aberdeen, Aberdeen AB24 3UF, United Kingdom

*Correspondence to: xi_chen@tju.edu.cn

**Abstract**

Conceptualising passive storage in coupled flow-isotope models can improve simulation of mixing and attenuation effects on tracer transport in many natural systems, such as catchments or rivers. However, the effectiveness of incorporating different conceptualisations of passive storage in models of complex karst flow systems remains poorly understood. In this study, we developed a coupled flow-isotope model that conceptualises both "fast" and "slow" flow processes in heterogeneous aquifers, in addition to hydrological connections between steep hillslopes and low-lying depression units in cockpit karst landscapes. The model tested contrasting configurations of passive storage in the fast and slow flow system and was optimized using a multi-objective optimization algorithm based on detailed observational data of discharge and isotope dynamics in the Chenqi catchment in southwest China. Results show that 1-3 passive storage zones distributed in hillslope fast/slow flow reservoir and/or depression slow flow reservoir provided optimal model structures in the study catchment. This optimization can effectively improve simulation accuracies for outlet discharge and isotope signatures. Additionally, the optimal tracer-aided model reflects dominant flow paths and connections of the hillslope and depression units, yielding reasonable source area apportionment for dominant hydrological components (e.g. more than ~80% of fast flow in the total discharge) and the solute transport in steep hillslope unit of karst flow systems. Our coupled flow-isotope model for karst systems provides a novel, flexible tool for more realistic catchment conceptualizations that can easily be transferred to other cockpit karst catchments.

**Keywords:** Flow-isotope model; passive storage; karst flow systems; Chenqi

catchment; Hillslope and depression units

## 1 Introduction

Karst areas cover extensive areas of the Earth's surface providing important water resources. For example, the southwest China karst region is one of the world's largest continuous karst areas, covering ~540 $\times$ 10$^3$ km$^2$ over eight provinces and providing water resources for more than 100 million people (Chen et al., 2018). The strong dissolution of carbonate rocks in the humid tropics and subtropics of southwest China creates unique cockpit karst landscapes, covering an area of about 140,000 - 160,000 km$^2$. Such cockpit karst morphology also occurs in areas in Southeast Asia, Central America and the Caribbean. In polje/tower karst systems, depression areas are interconnected with isolated towers scattered throughout the terrain (Lyew et al., 2007). Since hillslope runoff is regarded as a "water tower" often supplying agriculture in the depression, the development of hydrological models representing the hillslope and depression hydrological functionality is a necessary prerequisite for water resources management in cockpit karst landscapes.

A wide range of hydrological models have been developed for karst areas, ranging from lumped models at the catchment scale to (semi-) distributed models with hydrological function parameterized for grid-scales or landscape unit scales (Martínez‐Santos and Andreu, 2010; Hartmann et al., 2013; Husic et al., 2019; Dubois et al., 2020; Ollivier et al., 2020; Xu et al., 2020; Jeannin et al., 2021; Wunsch et al., 2022). A key function of karst hydrological models is to capture the dual or multi-phase flows in a complex porous medium, capturing low velocities in the matrix and small fractures, as well as very high velocities in large fractures and conduits (White, 2007;

Worthington, 2009; Jourde et al., 2018; Ding et al., 2020). Model structures endowed with process-based conceptualization of complex distributed flow systems often lead to over-parameterization and large uncertainties for resulting simulation (Perrin et al., 2001; Beven, 2006; Adinehvand et al., 2017). More generally, in recent years isotope-aided hydrological models have been developed to fully couple hydrological processes with stable isotope dynamics (Birkel and Soulsby, 2015). These coupled models are effective in quantifying hydrological functions, such as water storage, flux, and ages (Long and Putnam, 2004; Carey and Quinton, 2004; Delavau et al., 2017; Chacha et al., 2018; Zhang et al., 2020b; Elghawi et al., 2021; Mayer-Anhalt et al., 2022), which are useful metrics to characterize the karst critical zone and associated flow systems.

In isotope-aided hydrological models, flow routing is driven by pressure gradients, creating a dynamic (active) water storage that is influenced by water balance considerations (Fenicia et al., 2010; Soulsby et al., 2011), while tracer mixing, attenuation and transport require additional storage volumes (passive storage), such as unsaturated storage below field capacity (Birkel et al., 2011b) or saturated storage at depths far below the stream or water table. The conceptual combination of active storage with passive storage in isotope-aided hydrological models enhances solute mixing and resultant tracer retardation. As summarized in Table 1, previous tracer-aided hydrological models incorporate at least one passive storage. Generally, the number of passive storages increases with the sub-division of storage according to landscape units. For example, simple models with one (unsaturated/saturated or total)

storage unit have one passive storage parameter (Barnes and Bonell., 1996; Fenicia et al., 2010; Ala-Aho et al., 2017). For more complex models with at least two geographical units of uplands and lowlands, the number of passive storages could increase to 2-5 (Birkel et al., 2011a; Capell et al., 2012; Birkel et al., 2015; Mayer-Anhalt et al., 2022). Although these studies have provided a useful proof of concept, assessment of alternative configurations of passive storage functions has rarely been systematically tested.

For the complex flow systems in cockpit karst landscapes, a few studies have recently incorporated passive storage into coupled flow-isotope models for simulating hydrological and solute transport processes. For example, Zhang et al. (2019) developed a semi-distributed conceptual model for capturing discharge and isotope dynamics in the Chenqi catchment. The model has a function for passive storage to affect isotope mixing only within a conceptual hillslope unit, but it did not incorporate any passive storages in fast and slow reservoirs in the depression unit. Chang et al. (2020) compared lumped model structures with different connections of epikarst and the underlying slow and fast reservoirs according to observations of spring discharge and electrical conductivity (EC) in the Yaji catchment of southwest China. They set a passive storage for the fast flow reservoir but neglected passive storage in the slow flow system. These previous model structures with only one passive storage (Zhang et al., 2019; Chang et al., 2020) may not always be sufficient to simulate the distributed functioning of chemical mixing between active and passive storages and the hillslope

flow-depression flow inter-connections. Moreover, previous coupled models (listed in

Table 1) are mostly calibrated and validated only against daily and/or weekly

streamflow and isotope signatures. In karst catchments, as discharge responses and

isotope concentrations can vary extremely rapidly, the coarse resolution field data

cannot capture the hydrological and isotopic dynamics.

The overall aim of this study is to evaluate the effectiveness of alternative ways of

incorporating passive storage into a generic coupled flow-isotope model for cockpit

karst landscapes. The specific objectives were to: (1) develop a model that characterizes

the functions of fast and slow flow paths from hillslope to depression units for water

and tracer transport in cockpit karst landscapes; (2) systematically test alternative

passive storage configurations into the generalized model structure using a multi-

objective optimization algorithm based on detailed observational data of discharge and

isotope dynamics in the Chenqi catchment of southwest China; and (3), identify the

most appropriate model structures that most efficiently describe the hydrological

functioning of the catchment in terms of simulating the stream flow and tracer responses.

**Table 1.** Summary of the previous studies that account for passive storages in

hydrological models using at least one isotopic tracer

| Scale | Model | Number of passive storages | Location of passive storages | Tracer | Function | References |
|---|---|---|---|---|---|---|
| 25 ha | Models with fast and slow flow reservoirs | 1 | One storage | D | A | Barnes and Bonell., 1996 |
| 3.5 km$^2$ | Chemical-mixing | 2 | Shallow and deep storages | Chloride | A and B | Page et al., 2007 |

| | dynamic TOPMODEL | | | | | |
|---|---|---|---|---|---|---|
| 23.6 km$^2$ | The multiple bucket model | 3 | Soil storage | D | A | Son et al., 2007 |
| 3.8 ha | The SoftModel$_i$ | 2 | Upper and lower hillslope storages | D | A | Fenicia et al., 2008 |
| 3.8 ha | Complete mixing and partial-mixing model | 1 | One storage | D | B | Fenicia et al., 2010 |
| 2.3 and 122 km$^2$ | Lunan-CIM (L-CIM) | 2-5 | 2 for upper and low storages in upper catchment, and 3 for upper, low and deep storages in lower catchment | D | A | Birkel et al., 2011a |
| 3.6 and 30.4 km$^2$ | SAM$^{dyn}$ model | 1 | The total catchment storages | $^{18}$O | C | Birkel et al., 2011b |
| 749 km$^2$ | The tracer-aided model | 4 | Shallow and deep storages for uplands and lowlands | D and alkalinity | A | Capell et al., 2012 |
| 1.4, 8 and 9.6 km$^2$ | The DYNAMIT (DYNAmic MIxing Tank) | 2 | Unsaturated zone and slow flow reservoir | Chloride | A and B | Hrachowitz et al., 2013 |
| 30 km$^2$ | Tracer-aided hydrological model for a wet Scottish upland catchment | 3 | Three storages (upper, lower and saturation areas) | $^{18}$O | B and C | Birkel et al, 2015 |
| 3.7 km$^2$ | Hydrochemical model of Upper Hafren | 2 | Shallow and groundwater storage | Chloride | A and B | Benettin et al., 2015 |
| 3.2 km$^2$ | The landscape-based dynamic model | 3 | Three storages (hillslope, groundwater, and saturation area) | D | B | Soulsby et al., 2015 |
| 3.2 km$^2$ | STARR (Spatially Distributed | 2 | Soil and groundwater storage | D | A, B and C | van Huijgevoort et al., 2016 |

| | | | | | | |
|---|---|---|---|---|---|---|
| | Tracer-Aided Rainfall-Runoff model) | | | | | |
| 3.2, 0.6 and 0.5 km$^2$ | STARR (Spatially Distributed Tracer-Aided Rainfall-Runoff model) | 1 | Soil storage | $^{18}$O | A and B | Ala-Aho et al., 2017 |
| 3.2 km$^2$ | STARR model for the humid tropics | 2 | Soil and groundwater storage | D | A and C | Dehaspe et al., 2018 |
| 10.2 ha | A conceptual catchment model | 2 | Shallow and groundwater storage | $^{18}$O | A, B and C | Rodriguez., 2018 |
| 1.25 km$^2$* | Tracer-aided hydrological model for karst | 1 | Hillslope storage | D | A, B and C | Zhang et al., 2019 |
| 7.8 km$^2$ | STARR (Spatially Distributed Tracer-Aided Rainfall-Runoff model) | 2 | Soil and groundwater storage | D | A, B and C | Piovano et al., 2019 |
| 3.2 km$^2$ | A tracer-aided hydrological model | 3 | Dynamic hillslope reservoir, dynamic riparian zone reservoir and groundwater reservoir | D | A | Neill et al., 2019 |
| 126 km$^2$ | A coupled, tracer-aided, conceptual rainfall-runoff model | 4 | Four storages (upper, lower and saturation areas and deep groundwater) | D and dissolved organic carbon | A and B | Birkel et al., 2020 |
| Spring* | Lumped Model for karst | 1 | Fast flow reservoir | EC | A | Chang et al., 2020 |
| 0.23, 0.5, 0.6, 3.2 and 7.8 km$^2$ | A spatially distributed tracer-aided hydrological | 1 | Soil storage | D and $^{18}$O | A, B and C | Piovano et al., 2020 |

| | model (STARR) | | | | | |
|---|---|---|---|---|---|---|
| 1.44 km$^2$ | The EcH$_2$O-iso Model | 1 | The extra groundwater storage | D and $^{18}$O | A, B and C | Yang et al., 2021 |
| 3.9 km$^2$ | A conceptual tracer-aided hydrological model | 3 | The upper, lower and groundwater storages | D | A and B | Mayer-Anhalt et al., 2022 |
| 0.9 km$^2$* | The coupled flow-isotope model for karst catchment | 2 | Slow and fast flow reservoirs in hillslope and depression units | D | A and C | This study |

Note: A represents that passive storage can help reproduce the main isotope dynamics and improve simulation accuracy; B represents that passive storage can help track flux, resident or transit time; C represents that passive storage can help estimate catchment storage. D is the abbreviation for deuterium. *refers to application in a karst catchment.

**2 Study area and data descriptions**

**2.1 Study area**

Chenqi is a small karst catchment located in the Puding Karst Ecohydrological Observation Station, Guizhou Province of southwest China (Fig. 1). Chenqi belongs to the subtropical monsoon climate zone with a mean annual temperature of 14.2℃, mean annual rainfall of 1140 mm, and mean annual humidity of 78%. Precipitation mainly occurs in the rainfall season (May-August), accounting for more than 80% of the annual amount. The catchment is a typical karst peak cluster landform where a central depression is surrounded by hillslopes with elevations ranging from 1340m to 1530m. Considering the distinct topographic features, the catchment is conveniently divided into two dominant geomorphic units: hillslope and depression, with an area of 0.73 and 0.17 km$^2$, respectively (Table 2). Due to the peak cluster depression landform, runoff

generated from hillslopes mostly flows into depression aquifer prior to contributing to streamflow at the catchment outlet (Zhang et al., 2019; Zhang et al., 2020a).

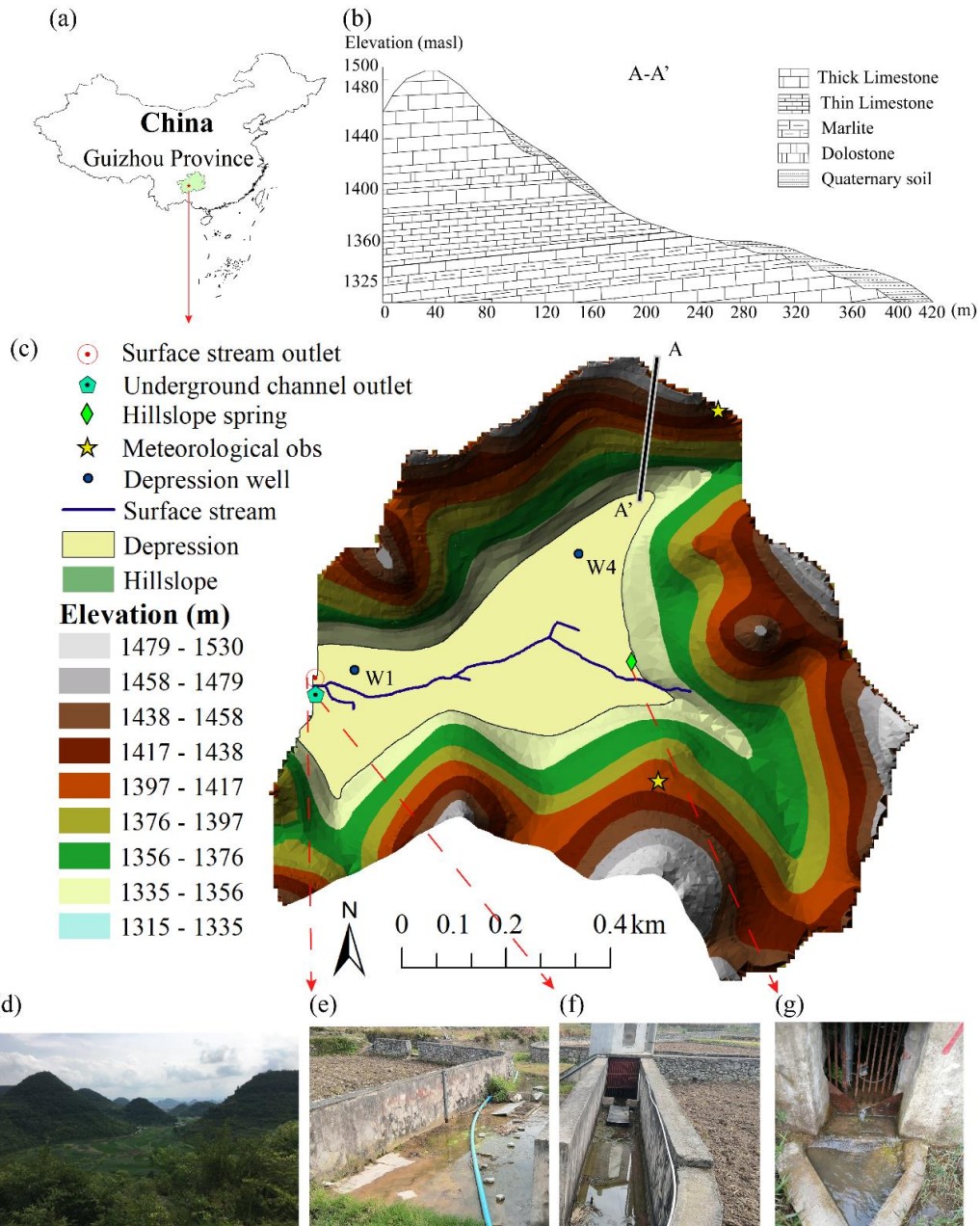

**Figure 1.** The location of Chenqi catchment (a), stratigraphic profile (b), topography (c), photo (d), and observations at surface stream outlet (e), underground channel outlet (f) and hillslope spring (g).

**Table 2**. The catchment characteristics of two landscape units at Chenqi

|  | Hillslope | Depression |
|---|---|---|
| Area (km²) | 0.73 | 0.17 |

| Elevation (m.a.s.l.) | 1340-1530 | 1315-1340 |
| Soil thickness | <0.5 m | >2 m |
| Land cover and use | Forest (13.67%), shrub (30.38%), grass (12.26%) and crops (40.1%) | |

## 2.2 Hydrogeological properties

Geological characteristics of the catchment include Quaternary soil, thick and thin limestone, dolostone, and marlite. The limestone formations with a thickness of 150-200 m lie above an impervious marlite formation (see A-A' profile in Fig. 1b) (Chen et al., 2018). In hillslopes, field investigations have shown a rich fracture zone (epikarst) which has a thickness of 7.5-12.6 m (Zhang et al., 2011). Quaternary soils consist of mostly sand (56-80%), fine sand (20-40%), calcareous soil and silt (1-10%). The soils are thin (less than 30 cm) and irregularly developed on carbonate rocks. Outcrops of carbonate rocks cover 10%-30% of the hillslope area. In some specific areas where a shallow impermeable layer (marlite) exists, hillslope springs appear on lower hillslopes. Deciduous broadleaved forests and shrubs are mostly grown on the upper and middle parts of hillslopes, and corn is grown at the foot of the gentle hillslopes (Chen et al., 2018).

In the low-lying depression, the accumulated soils are thicker (~2 m) and cultivated for crops of corn and rice paddy. The underlying limestone is strongly dissolved, producing underground conduits. These are sporadically distributed in the upper depression areas in connection with hillslope flows, and are gradually concentrated towards the catchment outlet (Cheng et al., 2019). The bedrock comprising the impervious marlite is located at depths of 30-50 m. Meanwhile, there are depression

ditches used for draining flood flow when the groundwater level is higher than the ditch

bottom (see surface stream in Fig. 1). So, the total outlet discharge is composed

predominantly of underground conduit flow in the study catchment, with surface

channel flow only in larger events.

**2.3 Observational dataset of hydrometry and stable isotope**

In the Chenqi catchment, an automatic meteorological station (Fig. 1c) was installed

to continuously record rainfall, temperature, air pressure, wind speed, humidity, and

solar radiation. These data were used to calculate the potential evaporation via the

Penman formula. Discharge at hillslope spring and the catchment outlet was measured

by v-notch weirs with a time interval of 15 min. All observational datasets were

collected from 8 October 2016 to 12 June 2018.

For stable isotope analysis, the hillslope spring, the catchment outlet flows, and

rainfall were sampled using an autosampler set. The sampling frequency was daily in

dry season (September - April) and hourly in the rainy season (May - August). In total,

we collected 253 rainfall samples, 1095 hillslope spring samples and 1096 water

samples at the catchment outlet of underground channel during the study period (Table

3).

As shown in Fig. 1c, there are two observation wells (W1 and W4) near the

catchment outlet and the upstream depression. W1 is located in a local confined aquifer,

consisting of extensively fractured carbonate rock surrounded by rock with low

secondary porosity. W4 is located in unconfined aquifer with the vertical permeability

reduction from large rock fracture and high secondary porosity to low secondary

porosity (Chen et al., 2018). The depression groundwater in the two wells (W1 and W4 in Fig. 1c) was manually sampled. Samples were taken at depths of 35 m and 13 m for W1 and W4, respectively, with a sampling frequency of two occasions before and after the four rainfall events from 6 July 2017 to 20 August 2017.

The sampled water was sealed by using plastic bags to avoid evaporation. Water samples were taken to the laboratory every day and stored at about 4 ℃. The water samples were tested and analyzed by the MAT 253 laser isotope analyser (instrument precision was ± 0.5 ‰ for δD and ± 0.1 ‰ for δ¹⁸O) at the State Key Laboratory of Hydrology and Water Resources of Hohai University.

**2.4 Characteristics of the observed hydrograph and stable isotope dynamics**

The observed surface, subsurface and total outlet flow (discharge) are shown in Fig. 2. The discharge response to rainfall is rapid, characterized by a sharp rise and decline of hydrographs. During the study period, the surface flow and underground flow are 43% and 57% of the total discharge, respectively. Various lines of evidence have demonstrated the hillslope-depression fast flow connection, particularly during heavy rainfall events. In the mid-season, after extremely heavy rainfall, hillslope spring discharge is highly synchronized with outlet flow, and the relationship between hillslope spring discharge and outlet discharge approaches a monotonic function (details in Zhang et al., 2020a). It is worth noting that due to the impact of agricultural irrigation, there were unreasonable sudden declines in surface and subsurface flow in June.

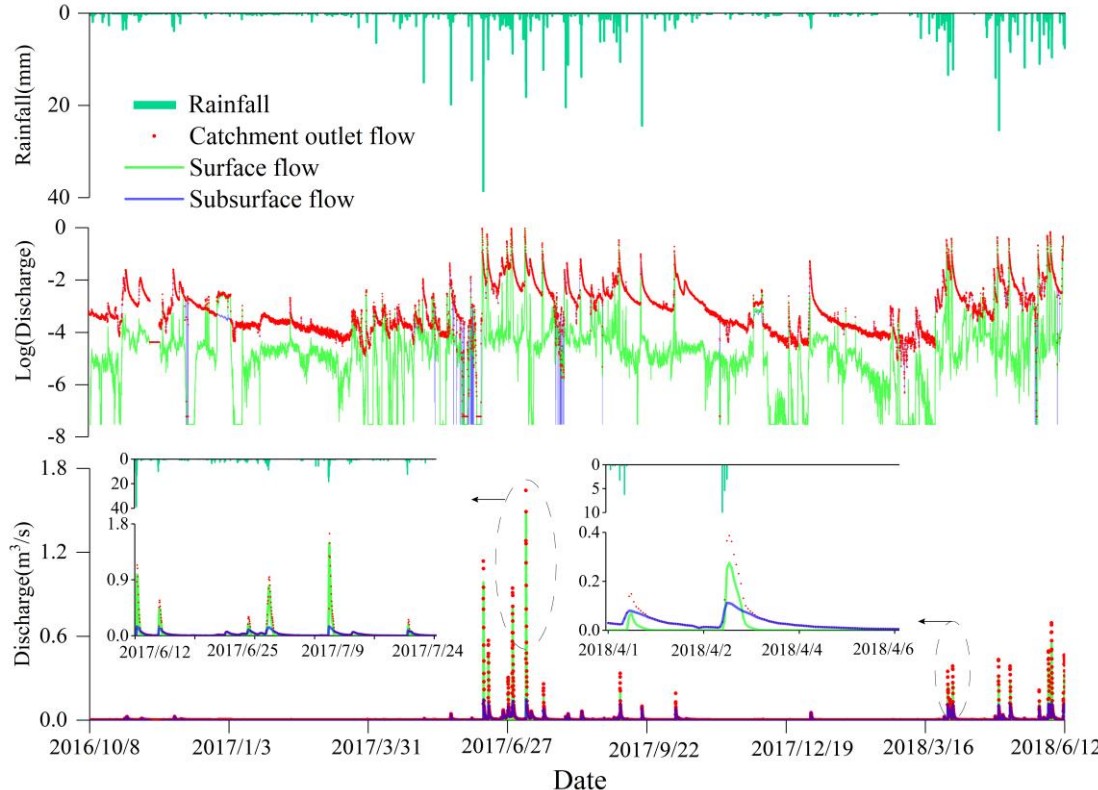

**Figure 2.** The observed surface, subsurface and catchment total outlet flow (discharge) during the study period.

The mean values of δD and δ$^{18}$O (Table 3) clearly show that the isotopic composition of water in the catchment becomes more enriched from the hillslope spring to the depression groundwater and the catchment outlet discharge. This implies increased mixing with more enriched groundwater affected by evaporative fractionation over the course of water flow paths from the hillslopes towards the outlet (Zhang et al., 2019; Zhang et al., 2020a). This is also illustrated by that the δD - δ$^{18}$O regression lines for hillslope spring and outlet discharge deviates from the LMWL (δD =8.18δ$^{18}$O+9.52) as shown in Fig. 3. Additionally, the regression line of hillslope flow is close to that of the catchment outlet discharge, inferring that hillslope flow is a primary source of the outlet discharge. The strong connection between hillslope flow and the outlet discharge is

attributed to widely spread of the high permeability zone in depression (e.g. at W4). The more depleted isotope signals at W4 show that groundwater there receives more new water (fast flow) from the hillslope spring and rainfall. By contrast, some older

water in the less permeable area of depression (e.g. at W1) still contributes to the outlet discharge. The more enriched $\delta^{18}$O and δD values at W1 show that flow there seldom mixes with new water (rainfall) (Chen et al., 2018), which could lead to a marked departure from the LMWL (Fig. 3).

The monthly statistical summaries of δD and lc-excess (lc-excess=δD-a·$\delta^{18}$O-β) are

shown in Fig. 4. In the wet season from May to October, the δD is gradually depleted, reflecting rainfall inputs, while in the dry season from November to April, the δD is gradually enriched. It indicates that both the hillslope spring and outlet discharge change from receiving more new rain water in wet season to being dominated by older water in the dry season. Meanwhile, the δD is more depleted and the lc-excess is more

positive for the hillslope flow, compared to the outlet discharge. It means that additional flow sources in the depression join the hillslope flow. This depression flow is older but undergoes less evaporation because of the flat topography and thicker soils. Nevertheless, the additional depression flow has little influence on discharge variability at the catchment outlet, as the various patterns of δD and lc-excess at catchment outlet

closely correspond to those of the hillslope spring.

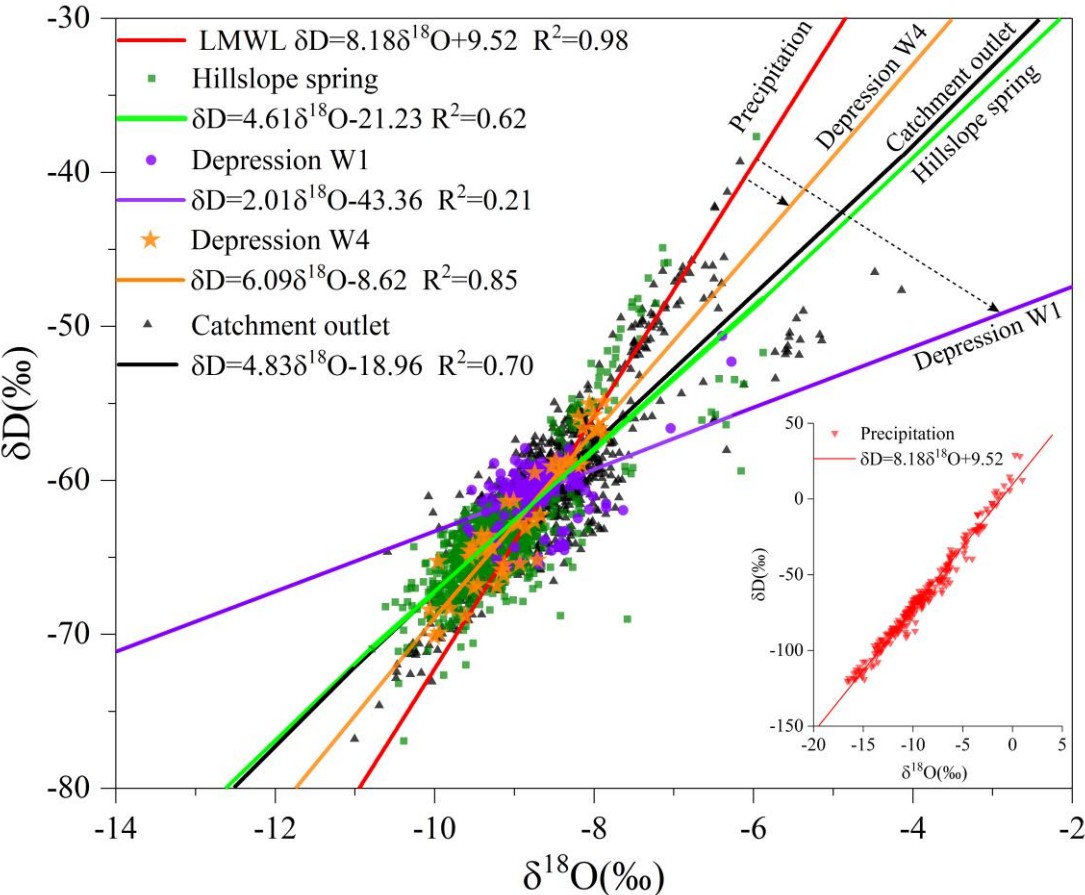

**Figure 3.** Plot of δ¹⁸O-δD for rainwater, catchment outlet discharge, hillslope spring and depression groundwater at wells W1 and W4. The correlation between $\delta^{18}O$ and $\delta D$ at W1 is 0.21, and tested to be significant at the significance level of $p<0.001$.

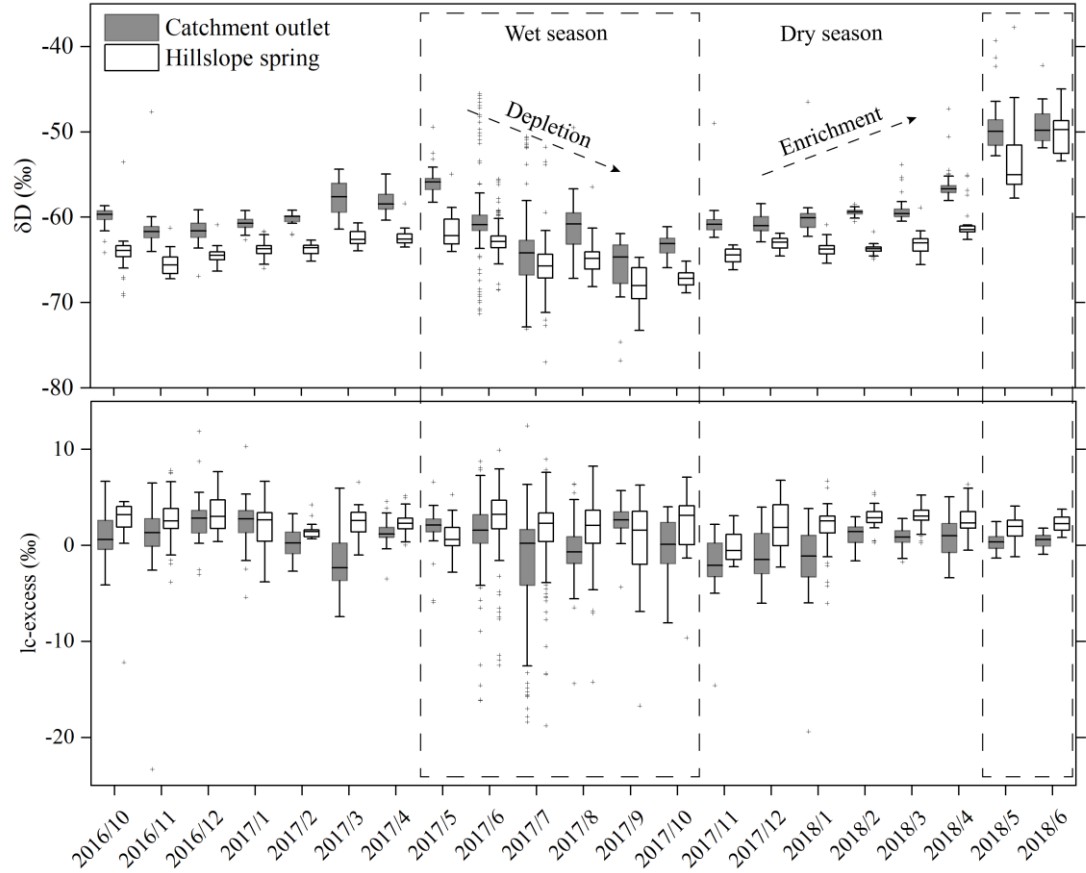

**Figure 4.** Monthly summaries of observed δD and lc-excess of outlet discharge and hillslope spring during the study period.

**Table 3.** Statistical characteristics of isotope data for rainfall, hillslope spring, catchment outlet discharge and depression groundwater in the study period

| Obs | Sampling time | Number | $\delta^{18}D$ (‰) Range | $\delta^{18}D$ (‰) Mean | $\delta^{18}O$ (‰) Range | $\delta^{18}O$ (‰) Mean | lc-excess Range | lc-excess Mean |
|---|---|---|---|---|---|---|---|---|
| Rainfall | | 253 | -120.2-29 | -64.9 | -16.6-1.0 | -9.1 | -16.71-17.37 | -0.04 |
| Catchment outlet discharge | Oct. 8 2016 - June 12 2018 | 1096 | -76.8- -39.3 | -60.6 | -11- -4.1 | -8.6 | -23.31-12.45 | 0.33 |
| Hillslope spring | | 1095 | -77- -37.8 | -63.7 | -10.8- -5.9 | -9.2 | -18.77-9.92 | 2.06 |
| Groundwater W1 | July 6- Aug. 20, 2017 | 175 | -65.7- -50.7 | -60.8 | -9.6- -6.3 | -8.7 | -10.75-7.6 | 0.65 |
| Groundwater W4 | July 6- Aug. 20, 2017 | 47 | -70.2- -55 | -62.5 | -10.1- -7.9 | -8.9 | -3.56-6.51 | 0.96 |

**3 Model development**

**3.1 Conceptual model structure**

Considering the contrasting features of the catchment landscape, the catchment area

is conveniently sub-divided into hillslope and depression units, and the model structure

can be conceptualized by focusing on the hydrologic connectivity of the "hillslope-

depression-stream" continuum (Zhang et al., 2020a). In each of hillslope and depression

units, the vertical profile is separated into an unsaturated zone comprising the soil and

epikarst layers, and a saturated zone representing the deep aquifer (Fig. 5). The effect

of spatial heterogeneity on hydrological functions is described by a distribution curve

of storage in the unsaturated, and a dual flow system in the saturated zone. The

distribution curve of storage, like a set of compartments in the VarKarst model

(Hartmann et al., 2013), has functions to quantify various recharge mechanisms (e.g.,

diffusive and concentrated allogenic and autogenic recharge). The dual flow system

consists of a fast flow reservoir and a slow flow reservoir that are interconnected and

can be used for groundwater routing (Hartmann et al., 2013; Zhang et al., 2019).

The steep hillslope flow moves to the low-lying depression with the following

possible connections: hillslope fast flow to depression fast/slow flow (HF-DF/DS), and

hillslope slow flow (HS) to depression fast/slow flow (Fig. 5). As hillslope fast flow is

primarily concentrated into depression conduits, the connection of HF-DS is neglected

in this study.

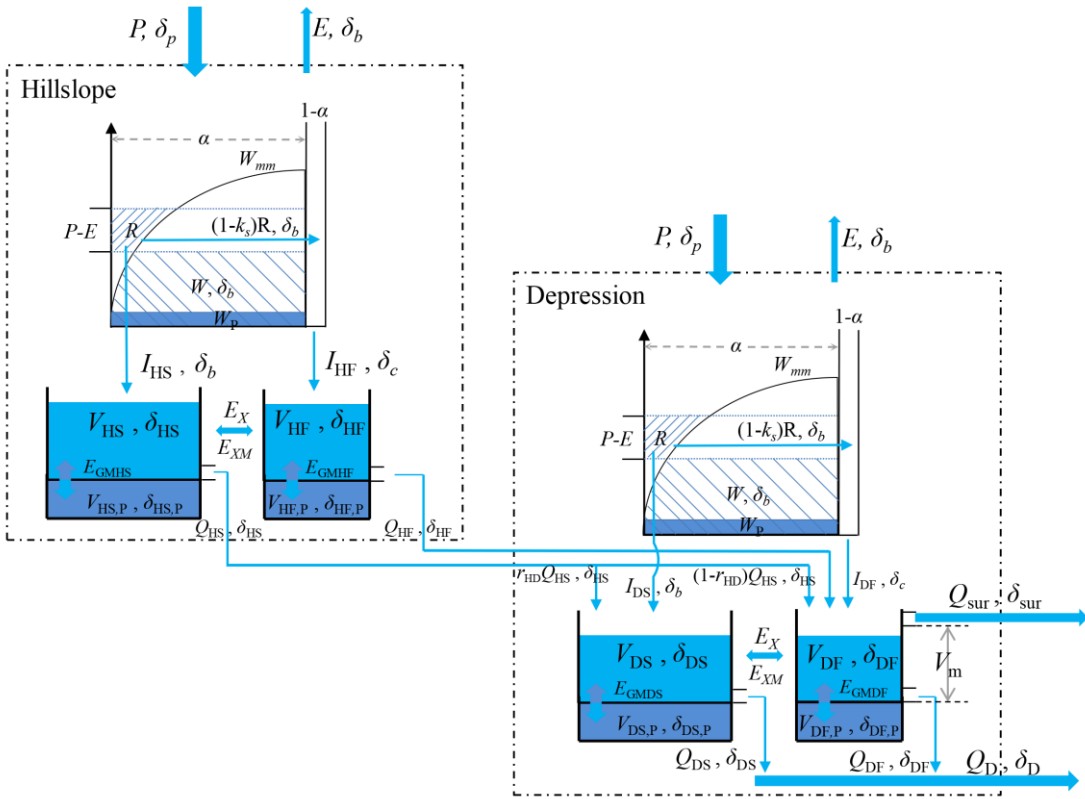

**Figure 5.** Conceptualized structure for the coupled flow-isotope model. The light blue shades indicate active storage, the dark blue shades indicate passive storage. The detailed descriptions of the model parameters are shown in Table 5.

### 3.1.1 Hydrological routing

In each of hillslope and depression units, the spatial heterogeneity of unsaturated storage volumes is described by a distribution curve of the storage capacity like the Xinanjiang model in Fig. 5 (Zhao, 1992) following:

$$\frac{f}{F} = 1 - (1 - \frac{W_{m'}}{W_{mm}})^b \tag{1}$$

where $f$ represents free water yield area, $F$ represents the total of the area ($\alpha$), $W_{m'}$ is the areal mean tension water storage at $f$, $W_{mm}$ is the maximum value of $W_{m'}$, and $b$ is a parameter.

Based on Eq. (1), the initial areal average storage $W$ is an integration of $W_{m'}$ within 0-$A$ in the area (1-f/F):

$$W = \int_0^A (1 - \frac{W_{m'}}{W_{mm}})^b dW_{m'} = \frac{W_{mm}}{1+b} \times \left[ (1 - \frac{A}{W_{mm}})^{1+b} \right] \qquad (2)$$

when $A = W_{mm}$, the storage in the entire area reaches the storage capacity. Thus, the mean storage capacity $W_m$ is equal to $\frac{W_{mm}}{1+b}$ (Zhao, 1992).

When the net precipitation $(P\text{-}E) > 0$ and if $P\text{-}E+A < W_{mm}$, the water yield $R$ is:

$$R = P - E - W_m + W + W_m \times (1 - \frac{P-E+A}{W_{mm}})^{1+b} \qquad (3)$$

Note that $P$ is precipitation and $E$ is actual evaporation estimated by $E = k_c \times E_P \times \frac{W}{W_{mm}}$,

in which $k_c$ is a coefficient for evapotranspiration, and $E_P$ is potential evapotranspiration.

If $P\text{-}E+A \geq W_{mm}$, the water yield $R$ is:

$$R = P - E - W_m + W \qquad (4)$$

The water yield $R$ recharges the deep aquifer, which is separated into diffusive recharge $I_S$ and concentrated allogenic and autogenic recharge $I_F$. The $I_S$ recharges the slow flow reservoir of the matrix or small fracture area with a ratio to hillslope or depression area of $\alpha$ (i.e., $I_S = k_s \times R \times \alpha$, where $k_s$ is the ratio of water yield into slow flow reservoir). The $I_F$ is the remaining runoff $((1\text{-}k_s) \times R \times \alpha)$ and rainfall $P$ falling on the swallow holes $(1\text{-}\alpha)$, which directly recharges fast flow reservoir (i.e., $I_F = P \times (1\text{-}\alpha) + R \times (1\text{-}k_s) \times \alpha$).

Consequently, in the saturated zone, the water balance in the fast and slow reservoirs is:

$$\frac{dV_S}{dt} = I_S - E_X - Q_S \qquad (5)$$

$$\frac{dV_F}{dt} = I_F + E_X - Q_F \qquad (6)$$

where $V_S$ and $V_F$ are storages of the slow and fast flow reservoirs, respectively; $Q_S$ and $Q_F$ are discharges from the slow and fast reservoirs, respectively; $E_X$ is flux between fast flow and slow flow reservoirs.

$E_X$ is estimated by difference of the saturated storages (or water heads) between the fast flow and slow flow reservoirs (i.e., $E_X = k_e \times (V_S - V_F)$), where $k_e$ is a coefficient of exchange flux between the slow and fast flow reservoirs). $Q_S$ and $Q_F$ are estimated according to the linear relationship between storage $V$ and discharge (i.e., $Q_S = \eta_S \times V_S$, and $Q_F = \eta_F \times V_F$, where $\eta_S$ and $\eta_F$ are outflow coefficients of slow and fast flow reservoirs, respectively).

### 3.1.2 Isotopic tracer routing

In each of the hillslope and depression units, the isotope mass balance in the unsaturated zone storage can be expressed as:

$$\frac{d(W_U \times \delta_b)}{dt} = P \times \delta_p - R \times \delta_b - E \times (1 + l_s) \times \delta_b \tag{7}$$

where $W_U$ ($W_U = W + W_P$) is the moisture storage consisting of active storage $W$ or mobile water (Sprenger et al., 2017; Sprenger et al., 2018) and passive storage $W_P$, $l_s$ is the coefficient of evaporation fractionation, $\delta_p$ and $\delta_b$ are the stable isotope ratios of rainwater ($P$) and moisture (and water yield $R$), respectively. Eq. (7) assumes instantaneous mixing of rainwater ($P$), water yield ($R$) and soil moisture ($W$), and complete mixing of the active storage ($W$) with passive storage ($W_P$) in the area ($\alpha$) since soils are very thin.

For the deeper aquifer, the mass balance in the slow and fast flow reservoirs is given by:

$$\frac{d(V_S \times \delta_S)}{dt} = I_S \times \delta_b - E_{XM} - E_{GMS} - Q_S \times \delta_S \tag{8}$$

$$\frac{d(V_F \times \delta_F)}{dt} = I_F \times \delta_c + E_{XM} - E_{GMF} - Q_F \times \delta_F \tag{9}$$

where $E_{XM}$ is the exchange mass between the slow flow and fast flow reservoirs (estimated by $k_e \times (V_S - V_F) \times \delta_S$ for $E_{XM} > 0$, and $k_e \times (V_S - V_F) \times \delta_F$ for $E_{XM} <= 0$); $E_{GMS}$ and $E_{GMF}$ represent the mixing of the solute between the active and passive storages for the slow and fast flow reservoirs, respectively; $\delta_S$ and $\delta_F$ are the stable isotope $\delta$ of the slow flow and fast flow, respectively.

Since $I_F$ comes from percolation of both unsaturated zone and direct rainfall recharge, the recharge water mass $I_F \times \delta_c$ is equal to

$$I_F \times \delta_c = P \times \delta_P \times (1 - \alpha) + \delta_b \times R \times (1 - k_s) \times \alpha \tag{10}$$

The mass balance of the passive storage ($V_P \times \delta$) affected by $E_{GMS}$ and $E_{GMF}$ for slow and fast flow reservoirs is:

$$\frac{d(V_{S,P} \times \delta_{S,P})}{dt} = E_{GMS} \tag{11}$$

$$\frac{d(V_{F,P} \times \delta_{F,P})}{dt} = E_{GMF} \tag{12}$$

where $V_{S,P}$ and $V_{F,P}$ are the passive storage of slow flow and fast flow reservoirs, respectively; $\delta_{S,P}$ and $\delta_{F,P}$ are the stable isotope $\delta$ of passive storage for the slow flow and fast flow reservoirs, respectively; $E_{GMS} = \varphi_S \times V_S \times (\delta_S - \delta_{S,P})$ and $E_{GMF} = \varphi_F \times V_F \times (\delta_F - \delta_{F,P})$, where $\varphi_S$ and $\varphi_F$ are the exchange coefficient between the active and passive storages for slow flow and fast flow, respectively.

The above Eqs. (8) and (11) describe partial mixing between $V_S$ and $V_{S,P}$ for the slow flow reservoir, and Eqs. (9) and (12) describe partial mixing between $V_F$ and $V_{F,P}$ for

the fast flow reservoir. Moreover, the partial mixing could be static or dynamic depending on whether the exchange coefficients between active and passive storages ($\varphi_S$ and $\varphi_F$) are constant or vary over time, respectively (Hrachowitz et al., 2013).

**3.1.3 Hillslope - depression connectivity and schematic model structures incorporating passive storage**

The hillslope fast flow is assumed to fully connect with fast pathways in depression (i.e., HF-DF in Table 4) while the hillslope slow flow passes through both the slow matrix and fast pathways in the depression (i.e., HF-DF/DS in Table 4). Therefore, the storages of $V_S$ and $V_F$ in the depression unit receive additional recharge from the hillslope slow flow. So, the hillslope slow/fast flow contribute to the depression slow/fast flow is $r_{HD} \times \dfrac{A_H}{A_D} \times Q_S$ and $(1-r_{HD}) \times \dfrac{A_H}{A_D} \times Q_S + \dfrac{A_H}{A_D} \times Q_F$, respectively, where $r_{HD}$ is a ratio of hillslope slow flow into depression slow flow, $A_H$ and $A_D$ are hillslope and depression areas, respectively. Correspondingly, $V_S \times \delta_S$ and $V_F \times \delta_F$ in the depression are influenced by the isotope composition of the hillslope inputs ($r_{HD} \times \dfrac{A_H}{A_D} \times Q_S \times \delta_S$ and

$(1-r_{HD}) \times \dfrac{A_H}{A_D} \times Q_S \times \delta_S + \dfrac{A_H}{A_D} \times Q_F \times \delta_F$ from the hillslope slow flow and fast flow, respectively).

There is a dual drainage system comprising both a surface stream and underground channel in the depression. Here, we set a critical volume $V_m$ in the depression. The catchment flow drains from surface stream $Q_{sur}$ only when the depression groundwater storage meets: $V_{DF} > V_m$ (i.e., $Q_{sur} = \dfrac{(V_{DF} - V_m) \times A_D}{\Delta t}$). As a consequence, the total flow

discharge at the catchment outlet $Q$ is composed of fast flow ($Q_F$) and slow flow ($Q_S$) in the subsurface, with additional contribution from the surface stream $Q_{sur}$.

The passive storage may exist in any flow systems (fast and slow flow) and geographical units (hillslope and depression) in karst catchments (Fig. 5). To optimize the number and positions of passive storage in the flow system, we set fourteen schemes (scenarios) that incorporates 0-4 passive storages into different positions of fast and slow reservoirs for hillslope and depression units (indicated by the subscript P in Table 4). The model parameters and their definitions are listed in Table 5.

**Table 4**. Different model structures that incorporate passive storages into fast flow and/or slow flow reservoirs at hillslope and/or depression units

| No. of Passive Storage | Model | Passive storage in hillslope | | Passive storage in depression | | Connection of flow system |
|---|---|---|---|---|---|---|
| | | Slow flow (HS) | Fast flow (HF) | Slow flow (DS) | Fast flow (DF) | |
| 0 | $a$ | - | - | - | - | HF-DF and HS-DS/DF |
| 1 | $b$ | P | - | - | - | HF-DF and HS$_P$-DS/DF |
| | $c$ | - | P | - | - | HF$_P$-DF and HS-DS/DF |
| | $d$ | - | - | P | - | HF-DF and HS-DS$_P$/DF |
| | $e$ | - | - | - | P | HF-DF$_P$ and HS-DS/DF$_P$ |
| 2 | $f$ | P | P | - | - | HF$_P$-DF and HS$_P$-DS/DF |
| | $g$ | - | - | P | P | HF-DF$_P$ and HS-DS$_P$/DF$_P$ |
| | $h$ | P | - | P | - | HF-DF and HS$_P$-DS$_P$/DF |
| | $i$ | - | P | - | P | HF$_P$-DF$_P$ and HS-DS/DF$_P$ |
| 3 | $j$ | P | P | P | - | HF$_P$-DF and HS$_P$-DS$_P$/DF |
| | $k$ | P | P | - | P | HF$_P$-DF$_P$ and HS$_P$-DS/DF$_P$ |
| | $l$ | - | P | P | P | HF$_P$-DF$_P$ and HS-DS$_P$/DF$_P$ |
| | $m$ | P | - | P | P | HF-DF$_P$ and HS$_P$-DS$_P$/DF$_P$ |
| 4 | $n$ | P | P | P | P | HF$_P$-DF$_P$ and HS$_P$-DS$_P$/DF$_P$ |

Note: The "P" and "-" represent the fast and slow flow reservoirs with and without passive storage, respectively.

## 3.2 Model calibration and validation


The observational data were used separately for the calibration and validation periods.

That is, the model parameters were calibrated against the observed discharge and

isotope concentration (δD) from October 8, 2016 to October 30, 2017. Afterwards, the

model was validated against observations from November 1, 2017 to June 12, 2018.

Note that since δD and $\delta^{18}O$ fluctuated with virtually the same dynamic over time and

both were driven by the same hydrological factors, therefore only δD was used for

calibration. The flow-isotope coupled models with different combinations of the active

and passive storages (Table 4) were run on hourly time steps.

In this study, the multi-objective optimization algorithm, i.e., non-dominated sorting

genetic algorithm II (NSGA-II) proposed by Deb et al. (2002), was applied for the

model parameter calibration. The NSGA-II algorithm (Kollat and Reed, 2006) based

on NSGA algorithm can identify the sets of pareto-optimal solutions. As pareto-optimal

sets of solutions are not dominated by any one of the factors as a result of trade-off

effects, the "best" solution is achieved by satisfying the demands from the performance

objective functions including the modified Kling-Gupta efficiency (KGE) and the

absolute value of BIAS ($Abias_q$) (Fenicia et al., 2007). KGE criterion comprehensively

considers the linear correlation and standard deviation between the numerical and

observed values (Kling et al., 2012) following:

$$KGE_i = 1 - \sqrt{(r-1)^2 + (std-1)^2 + (\mu-1)^2}$$  (13)

where $r$ is the linear correlation coefficient between the simulated and observed values, $std$ is the ratio of the standard deviation of the numerical and observed values, and $\mu$ is the ratio of the average numerical value to the observed value, i = (q, c) representing the goodness of match for flow discharge or isotope concentration, respectively. The closer KGE is to 1, the better the overall performance of the coupled model.

The $\text{Abias}_q$ is

$$\text{Abias}_q = | \frac{\sum\limits_{i=1}^{n}(S_i - O_i)}{\sum\limits_{i=1}^{n} O_i} | \qquad (14)$$

where $S_i$ is the simulated discharge, and $O_i$ is the observed discharge. The closer $\text{Abias}_q$ is to 0, the better performance of model in matching flow discharge at the outlet.

For a number of iterations (e.g. 1000 in this study), 50 parameter sets were initially 420 retained. Then the remaining sets with $\text{Abias}_q$ less than or equal to 0.2 in the 50 parameter sets, are sorted from the largest to the smallest according to the sum of corresponding $\text{KGE}_q$ and $\text{KGE}_c$. Finally, 30 sets are selected as the pareto-optimal solution (Nan et al., 2021). The corresponding objective function values (average of the optimal solution sets) for both the calibration and validation periods were extracted.

The range of each parameter value is initially set for model calibration according to our previous investigations (Zhang et al., 2019; Zhang et al., 2020a; Xue et al., 2019). The volumes of passive storages ($W_{H,P}$ and $W_{D,P}$; $V_{S,P}$ and $V_{F,P}$) are generally one order of magnitude larger than those of active storage (Dunn et al., 2010; Soulsby et al., 2011; Ala-Aho et al, 2017). So the ranges of $W_{H,P}$ and $W_{D,P}$ in the unsaturated zone are set as

 500-550mm, and the ranges of $V_{H,P}$ and $V_{D,P}$ in saturated zone are set as 300-350mm.

Considering the rapid hydrological response of the fast flow system or hillslope unit to

precipitation, the initial values of active storage ($V_{HF}$, $V_{DF}$ and $V_{HS}$) are set as 0 mm,

while the initial value of $V_{DS}$ is 20 mm (Xue et al., 2019). Meanwhile, the isotope ratios

for deuterium are all initially set to the measurement at the catchment outlet (i.e., -

 61.3‰), this initialisation brings negligible errors since isotope transport is driven by

rainfall inputs boundary condition.

A regional sensitivity analysis (Freer et al., 1996) was executed to identify the most

important model parameters. The sensitive parameters targeting $KGE_q$ are the ratio of

water yield into slow flow reservoir ($k_{sH}/k_{sD}$), the maximum storage of the fast flow

 reservoir $V_m$, and the outflow coefficient of fast flow reservoir in hillslope unit ($\eta_{HF}$).

There are other sensitive parameters when targeting on $KGE_c$, including $\alpha_H$, $k_{cH}$, $k_{sH}$, $b_H$,

$W_{mH}$ and $\eta_{HS}$ in the hillslope unit, and $\alpha_D$, $k_{cD}$, and $\eta_{DS}$ in the depression unit. Overall,

the parameters in the hillslope unit are more sensitive to discharge and isotopic ratios,

compared with those in the depression unit.


**Table 5.** The definitions of model parameters with their ranges

| Zone | Parameter and meaning | | Range |
|---|---|---|---|
| Area | $\alpha_H/\alpha_D$ | Ratio of matrix flow area | 0.90-0.95/0.95-1 |
| Unsaturated | $k_{cH}/k_{cD}$ | Coefficient for evapotranspiration | 0.9-1.3 |
| | $k_{sH}/k_{sD}$ | Ratio of water yield into slow flow reservoir | 0.1-0.5 |
| | $b_H/b_D$ | Exponential distribution of tension water capacity | 0.1-0.3 |
| | $l_{sH}/l_{sD}$ | coefficient of evaporation fractionation | 0-0.1 |
| | $W_{mH}/W_{mD}$ | Tension water storage capacity (mm) | 40-60/70-90 |

| | #$W_{H,P}$/$W_{D,P}$ | passive storage (mm) | 500-550 |
|---|---|---|---|
| | -/$Vm$ | Maximum storage of fast flow reservoir (mm) | 30-50 |
| | $r_{HD}$ | Ratio of hillslope slow flow into slow flow reservoir in depression | 0.1-0.8 |
| | $\eta_{HS}$/$\eta_{DS}$ | Outflow coefficient of slow flow reservoir | 0.001-0.01 |
| | $\eta_{HF}$/$\eta_{DF}$ | Outflow coefficient of fast flow reservoir | 0.01-0.15 |
| Saturated | $k_{eH}$/$k_{eD}$ | Exchange coefficient between slow and fast flow reservoirs ($10^{-3}$) | 0.1-1 |
| | #$\varphi_{HS}$/$\varphi_{DS}$ | Exchange coefficient between active and passive storages for slow flow | 0.1-0.5 |
| | #$\varphi_{HF}$/$\varphi_{DF}$ | Exchange coefficient between active and passive storages for fast flow | |
| | #$V_{HS,P}$/$V_{DS,P}$ | Passive storage for slow flow (mm) | 300-350 |
| | #$V_{HF,P}$/$V_{DF,P}$ | Passive storage for fast flow (mm) | |

Note: the upper and lower parameters and values in "*/*" represent those in hillslope and depression, respectively; the parameters indicated by "#" refer to those used for isotope concentration simulation. "-" represents not available.

## 4 Results

### 4.1 Performance of models during calibration and validation periods

The 30 optimal solutions and their means for the objective functions of $KGE_q$, $KGE_c$ and $Abias_q$ are obtained from parameter calibration of 14 models as shown in Table 6 and Fig. 6. Most models obtain a higher $KGE_q$ but a lower $KGE_c$, which was also reported by other studies (Soulsby et al., 2015; Dehaspe et al., 2018; Mudarra et al., 2019; Birkel et al., 2020). For the models incorporating 0-4 passive storages in Table 4, the accuracy of the simulated discharge and isotopic concentration does not increase with the number of passive stores. Comparatively, models $c$, $f$ and $j$ give higher mean values for both $KGE_q$ (>0.65) and $KGE_c$ (>0.55) (Table 6), and the models $c$ and $f$ also obtain a more constrained range of $KGE_q$ and $KGE_c$ from the 30 sets of optimal solutions (Fig. 6) in the calibration and validation periods. All of the three better

performing models have a passive storage in the hillslope fast reservoir but do not incorporate any passive storage in the depression fast reservoir (see Table 4). This indicates that hillslope (fast) flow and isotope mixing catchment outlet discharge and isotopic concentration, are consistent with the inferences from the observational data analysis.

As an example, Figs. 7 and 8 show the outlet discharge and isotope ($\delta$D) variations, respectively, simulated by model $f$. Model $f$ can generally capture the flood peaks (Fig. 7) and the isotope ($\delta$D) variations (Fig. 8). The average $KGE_q$ and $KGE_c$ from model $f$ are higher than 0.59 in the calibration and validation periods, and $Abias_q$ is relatively small (Table 6). Fig. 9 shows that $KGE_q$ is negatively correlated with $KGE_c$ according to the 30 optimal solution sets by the NSGA-II algorithm. Therefore, the multi-objective calibration gives a trade-off solution pair of high values for both $KGE_q$ and $KGE_c$ for the calibrated model $f$ as well as models $c$ and $j$. The other models do not balance the trade-off between $KGE_c$ against $KGE_q$ as effectively. For example, model $n$ with four passive storages obtains high $KGE_q$ (>0.6) but low $KGE_q$ (<0.3) (Table 6).

**Table 6.** Model performance based on the average of 30 optimal solution sets for each individual model structure

| No. of Passive Storage | Model | Calibration | | | Validation | | |
|---|---|---|---|---|---|---|---|
| | | $KGE_q$ | $KGE_c$ | $Abias_q$ | $KGE_q$ | $KGE_c$ | $Abias_q$ |
| 0 | $a$ | 0.46 | 0.30 | 0.08 | 0.46 | 0.38 | 0.23 |
| 1 | $b$ | 0.54 | 0.24 | 0.07 | 0.52 | 0.51 | 0.22 |
| | $c$ | **0.65** | **0.61** | **0.08** | **0.68** | **0.73** | **0.16** |
| | $d$ | 0.42 | 0.31 | 0.09 | 0.4 | 0.04 | 0.25 |
| | $e$ | 0.52 | 0.45 | 0.09 | 0.53 | 0.22 | 0.18 |

| | | | | | | | |
|---|---|---|---|---|---|---|---|
| 2 | **f** | **0.68** | **0.59** | **0.09** | **0.72** | **0.73** | **0.14** |
| | g | 0.47 | 0.39 | 0.1 | 0.48 | -0.12 | 0.19 |
| | h | 0.52 | 0.32 | 0.08 | 0.5 | 0.29 | 0.23 |
| | i | 0.65 | 0.15 | 0.07 | 0.67 | 0.5 | 0.12 |
| 3 | **j** | **0.66** | **0.55** | **0.09** | **0.67** | **0.72** | **0.16** |
| | k | 0.66 | 0.24 | 0.1 | 0.68 | 0.59 | 0.16 |
| | l | 0.63 | 0.21 | 0.08 | 0.64 | 0.32 | 0.14 |
| | m | 0.52 | 0.42 | 0.08 | 0.53 | 0.11 | 0.19 |
| 4 | n | 0.62 | 0.22 | 0.1 | 0.61 | 0.29 | 0.16 |


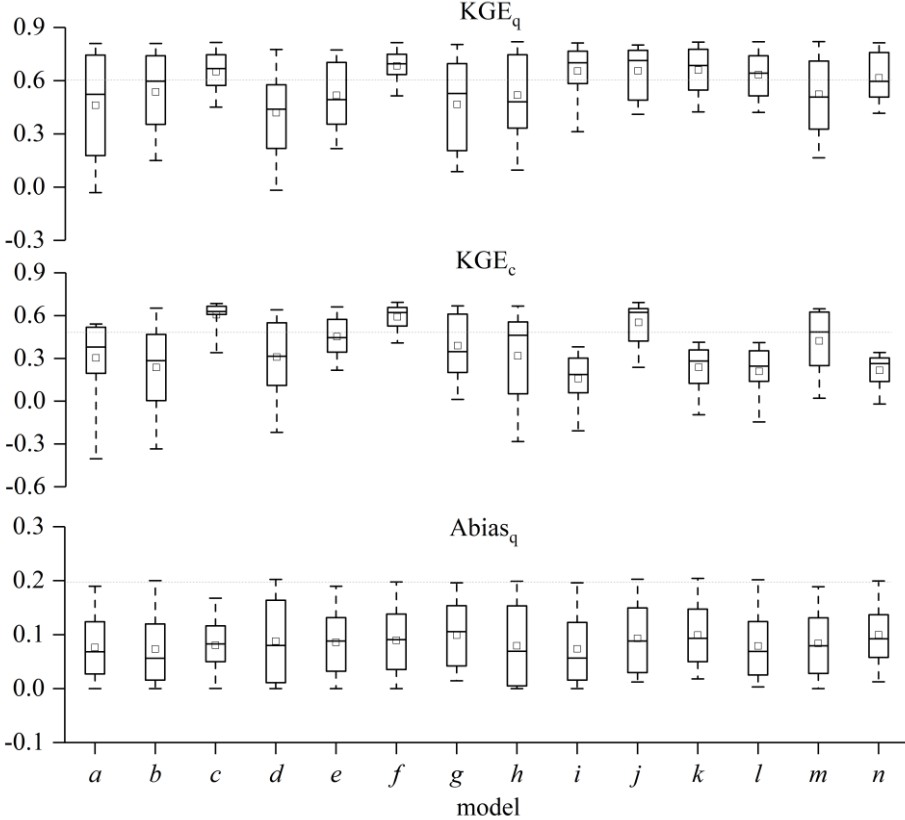

**Figure 6**. The box-plot of the 30 optimal solutions for the objective functions of $KGE_q$, $KGE_c$ and $Abias_q$ obtained from parameter calibration of 14 models.

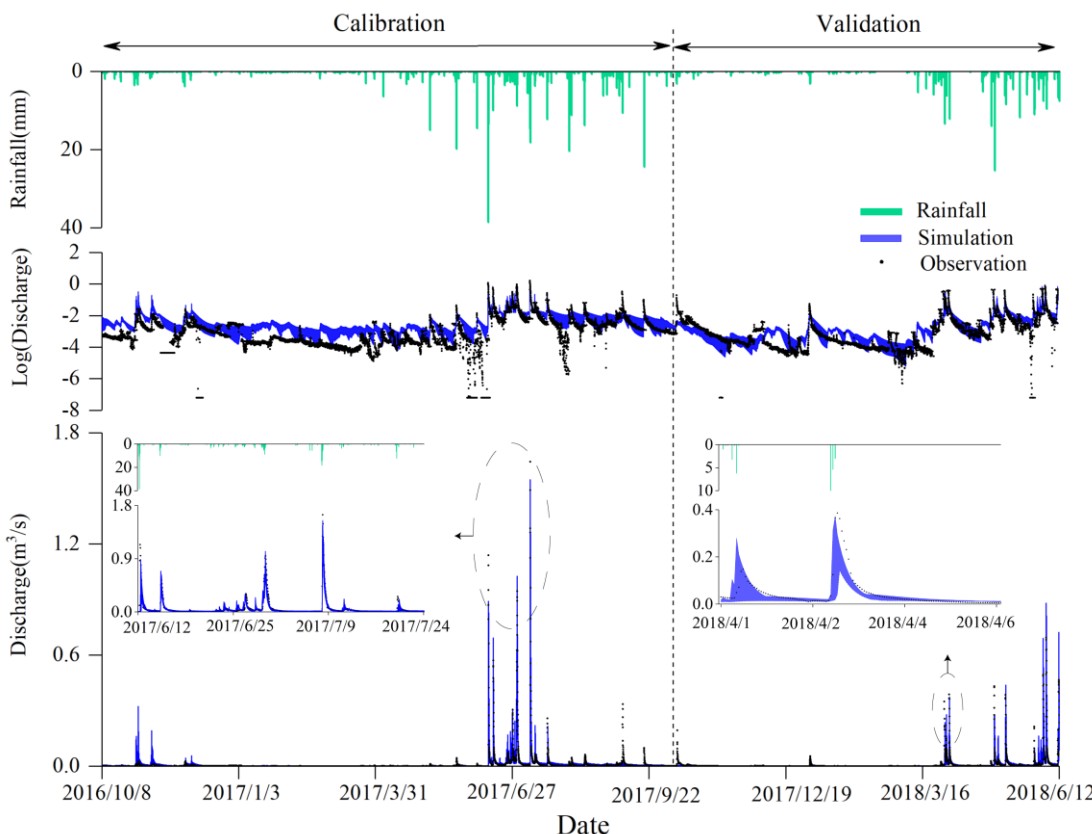

**Figure 7**. Simulated discharge concentrations of the 30 sets of optimal solutions by model *f* in calibration and validation periods. Note: The blue shades represent the simulated range of the 30 optimal solution sets; the black dots represent the observed discharge (the total of surface and subsurface discharge) at the catchment outlet.

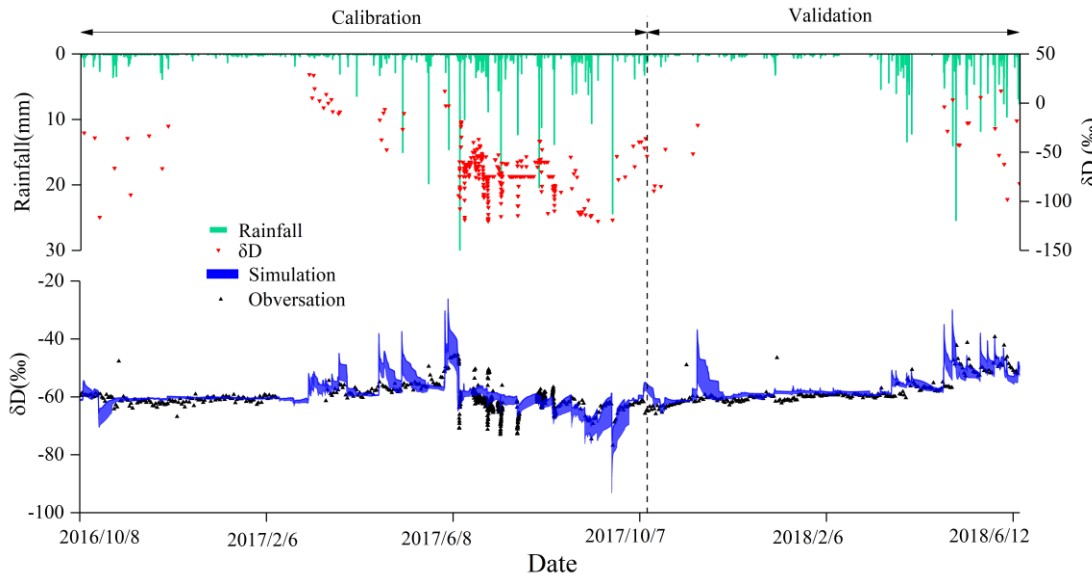

**Figure 8**. Simulated isotope concentrations of the 30 sets of optimal solutions by model *f* in calibration and validation periods. Note: The blue shades represent the simulated range of the 30 optimal solution sets; the black dots represent the observed isotope concentrations at the catchment outlet.

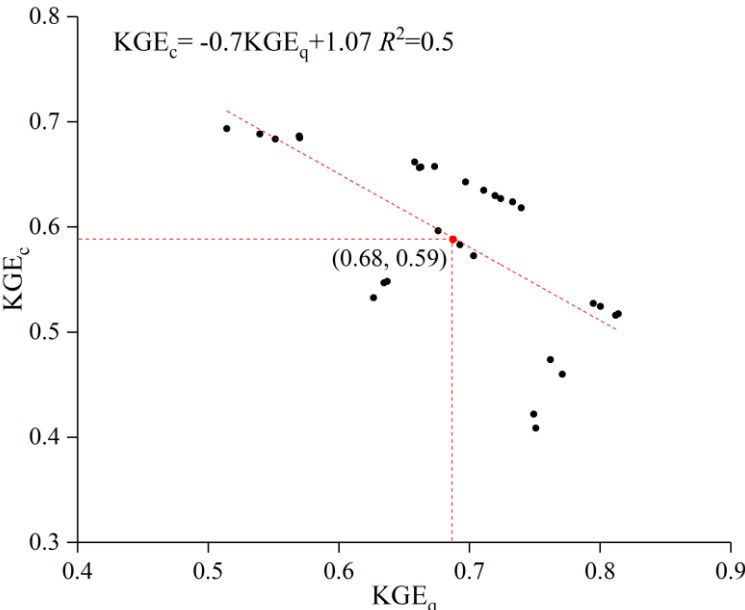

**Figure 9**. Relationship between $KGE_q$ and $KGE_c$ from the 30 optimal solution sets of model $f$.

## 4.2 Calibrated parameter values

The calibrated parameter values for the better performing models $c$, $f$ and $j$ are listed

in Table 7. These parameter values reasonably delineate the hydrological features of

karst landforms. For example, the calibrated $k_s$ ranges 0.13 - 0.24 in hillslope and

depression units, suggesting about 76-87% of net precipitation recharging into fast flow

reservoir through large fracture and sinkhole in terms of $I_f/R=(1-\alpha)P/R+(1-k_s)\alpha$. This

high percentage is consistent with the numerical results of Zhang et al. (2011)

independently derived using a distributed model that takes account of the role of

sinkholes in facilitating fast flow recharge into the aquifer in the studied catchment.

Charlier et al.(2012) found that about 60% of recharge water entered the conduit

network (fast channelized flow paths) in a small karst system in the French Jura

mountains. Worthington et al. (2000) also revealed that more than 90% of fast flow

component in four typical karst aquifers in Kentucky, USA. $W_m$ representing the soil

moisture retention capacity ranges 52-58 mm for thin soils over hillslope, substantially smaller than 81-90 mm for thick soils over depression according to the calibrated results of the three better models. The outflow coefficient of the fast flow reservoir $\eta_F$ (0.14-0.15/0.01-0.02 for the hillslope/depression) is much greater than that of slow flow reservoir $\eta_S$ (0.002-0.004/0.003-0.005), especially for the hillslope unit. This suggests that fast flow discharge is much more sensitive to active storage variability than slow flow discharge since $Q=\eta_S \times V$. In addition, the optimized ratio of hillslope slow flow contribution to depression slow flow $r_{HD}$ is close for models $c$ and $f$ (0.37 and 0.39, respectively), which are smaller than 0.55 for model $j$. The larger $r_{HD}$ value for model $j$ means more hillslope slow flow allocation to the depression slow flow reservoir.

**Table 7.** The mean values of model parameters for the 30 optimal solution sets from the three better models

| Zone | Parameter | Model $c$ | Model $f$ | Model $j$ |
|---|---|---|---|---|
| Area | $\alpha_H/\alpha_D$ | 0.92/0.99 | 0.94/0.99 | 0.94/0.98 |
| Unsaturated | $k_{cH}/k_{cD}$ | 1.14/1.08 | 1.12/1.04 | 1.17/1.15 |
| | $k_{sH}/k_{sD}$ | 0.24/0.13 | 0.22/0.14 | 0.16/0.23 |
| | $b_H/b_D$ | 0.14/0.24 | 0.11/0.15 | 0.24/0.15 |
| | $l_{sH}/l_{sD}$ | 0.01/0.01 | 0.01/0.05 | 0.02/0.02 |
| | $W_{mH}/W_{mD}$ | 58/90 | 56/82 | 52/81 |
| | #$W_{H,P}/W_{D,P}$ | 547/534 | 535/509 | 528/517 |
| Saturated | -/$Vm$ | 44 | 36 | 35 |
| | $r_{HD}$ | 0.37 | 0.39 | 0.55 |
| | $\eta_{HS}/\eta_{DS}$ | 0.002/0.005 | 0.004/0.003 | 0.003/0.004 |
| | $\eta_{HF}/\eta_{DF}$ | 0.15/0.01 | 0.14/0.01 | 0.14/0.02 |
| | $k_{eH}/k_{eD}$ | 0.2/0.3 | 0.2/0.3 | 0.3/0.5 |
| | #$\varphi_{HS}/\varphi_{DS}$ | -/- | 0.18/- | 0.22/0.29 |
| | #$\varphi_{HF}/\varphi_{DF}$ | 0.25/- | 0.26/- | 0.19/- |
| | #$V_{HS.P}/V_{DS.P}$ | -/- | 316/- | 331/323 |
| | #$V_{HF.P}/V_{DF.P}$ | 322/- | 325/- | 334/- |

Note: the upper and lower parameters and values in "*/*" represent those in hillslope and depression, respectively; the parameters indicated by "#" refer to those used for isotope concentration simulation. "-" represents not available in the models.

**4.3 The effects of passive storage on simulated flow composition and isotopic concentration**

We further compared the simulated flow components and their isotopic concentrations of the three better performing models (model *c*, *f* and *j* with passive stores 1-3 in Table 6). The results of model *a* without any passive store are also used as a benchmark for comparison. The partitioning of simulated outlet discharge by the four models is listed in Table 8. All three better models with passive stores set in the hillslope unit have a high proportion of discharge from the fast flow system, particularly in the hillslope unit. In the hillslope unit, model *a* obtains 79% of the fast flow component and 21% of the slow flow component, while the three better models with passive stores in the hillslope give a higher proportion of discharge from the fast flow system (87%). In the depression unit and catchment outlet, the simulated slow flow composition is slightly different, while the simulated proportions of the underground fast flow and surface flow are largely different in the three models. As shown in Table 8, model *f* gives 44% of surface stream flow and 56% of underground channel flow (the total of fast and slow flow), which are close to observed values at the surface stream (43%) and underground channel (57%) outlets. By contrast, models *a*, *c* and *j*, particularly model *a*, underestimate surface stream flow and overestimate underground channel flow.

The simulated isotope values of the flow components in the hillslope-depression-outlet continuum are listed in Table 9. Compared with model *a*, models *c*, *f* and *j* with passive storages increase isotope mixing and lead to a reduction of the δD variability (see the narrower range of δD for models *c*, *f* and *j* in Table 9). Meanwhile, as the

number of passive storages increases in the models, the mixing of fast flow and slow

flow is enhanced, leading to the mean δD values of slow flow approaching that of fast

flow. Nevertheless, for the three better models, the strengthened mixing of slow flow

only has a limited effect on the mean δD of fast flow as the mean δD of the catchment

outlet flow is closer to that of fast flow. It further supports the hypothesis that hillslope

fast flow dynamics control the catchment flow and isotopic concentration at the outlet.

**Table 8.** The proportions of flow components in the hillslope-depression-outlet
continuum for the 30 optimal solution sets of the selected representative models during
the study period (%)

| No. of Passive storage | Model | Hillslope | | | | Depression and catchment outlet | | | | | |
|---|---|---|---|---|---|---|---|---|---|---|---|
| | | Slow flow | | Fast flow | | Slow flow | | Fast flow | | Surface flow | |
| | | Range | Mean | Range | Mean | Range | Mean | Range | Mean | Range | Mean |
| 0 | a | 4-34 | 21 | 66-96 | 79 | 4-20 | 12 | 36-80 | 56 | 0-57 | 32 |
| 1 | c | 6-27 | 13 | 73-94 | 87 | 3-15 | 8 | 39-60 | 54 | 27-57 | 38 |
| 2 | f | 6-27 | 13 | 73-94 | 87 | 4-15 | 8 | 37-58 | 48 | 31-59 | 44 |
| 3 | j | 6-29 | 13 | 71-94 | 87 | 4-17 | 9 | 38-74 | 51 | 20-56 | 40 |

Note: the total flow at the catchment outlet is the sum of slow flow, fast flow and surface flow.

**Table 9.** The simulated isotope values (‰) of flow components in the hillslope-
depression-outlet continuum for the 30 optimal solution sets from the selected
representative models during the study period

| No. of Passive storage | Model | Hillslope | | | | Depression and catchment outlet | | | | | | | |
|---|---|---|---|---|---|---|---|---|---|---|---|---|---|
| | | Slow flow | | Fast flow | | Slow flow | | Fast flow/Surface flow | | Catchment outlet | | | |
| | | Range | Mean | Range | Mean | Range | Mean | Range | Mean | Range (Sim.) | Mean (Sim.) | Range (Obs.) | Mean (Obs.) |
| 0 | a | -65.1- -35.8 (29.3) | -55.2 | -102.6-9.8 (112.4) | -58.5 | -73- -37 (36) | -56.4 | -96.1- -8.9 (87.2) | -57.8 | -93.3- -9.8 (103.6) | -57.5 | | |
| 1 | c | -70.7- -39.3 (31.4) | -56.3 | -93-9.5 (102.5) | -59.4 | -72.3- -41.9 (30.4) | -57.8 | -84.3- -30.7 (53.6) | -59.4 | -79.4- -31.6 (47.8) | -59.2 | -76.8- -39.3 (37.5) | -60.6 |
| 2 | f | -63.4- -44.8 (18.6) | -59.4 | -95.2-10 (105.2) | -59.8 | -68.4- -43.3 (25.1) | -58.9 | -83.4- -31.5 (51.9) | -59.7 | -80.9- -32.8 (48.1) | -59.6 | | |
| 3 | j | -61.8- -39.4 (22.4) | -59.3 | -96.2-9.7 (105.9) | -59.5 | -62.7- -49.8 (12.9) | -60 | -85.1- -26.6 (58.5) | -59.2 | -84.7- -28.8 (55.9) | -59.2 | | |

Note: the number in blanket refers to the range of δD.

**5. Discussion**

**5.1 Importance of passive storage for tracer-aided hydrological modeling**

Involving passive storage for coupled flow-isotope model helped to improve the

performance of the discharge simulations, whilst being able to capture tracer dynamics,

which has been demonstrated by most previous studies (see Table 1). However, the

exact configuration of how passive storage can be set in different positions for different

models (Table 1), or even for a specific model. Taking the STARR model as an example,

van Huijgevoort et al. (2016), Dehaspe et al. (2018) and Piovano et al. (2019) added

two passive storages for the soil and groundwater stores, while Ala-Aho et al.(2017)

and Piovano et al. (2020) only used a passive storage in the soil store. Of all the studies

in Table 1, only Fenicia et al. (2008) and Birkel et al.(2011b) compared the simulation

effects of model structures on discharge with and without passive storages.

Most previous studies have focused on non-karst catchments, and passive storages

are usually represented in slow flow reservoirs (Hrachowitz et al., 2013; Yang et al.,

2021; see Table 1). Birkel et al. (2011a) and Hrachowitz et al. (2013) suggested that

this passive storage can be interpreted as soil moisture below field capacity or

groundwater below the dynamic storage. For more complex model structures,

delineating flow components and connections in heterogeneous landscape units usually

requires more flow routing compartments and thus additional passive storages. For

example, Capell et al. (2012) identified that only three passive storages were necessary

for a tracer-aided model with four possible passive storages in upland and lowland units

in the North Esk catchment in northeast Scotland. They found that passive storage in

shallow zone for the upland unit was negligible as sufficient damping was available in

the dynamic (active) storage.

Required model structures are usually more complex in karst catchments due to different conceptualisation of recharge and flow mechanisms. Most studies have demonstrated that the fast channelized flow paths control the sharp rise and decline of the hydrograph, and thus setting passive storage in fast flow reservoir can improve

simulation accuracy of the catchment flow and tracer dynamics in karst catchments, particularly in cockpit karst landscapes. For example, Zhang et al. (2019) assumed that hillslope flow is rapid, and showed that directly setting a passive storage in the hillslope flow reservoir can successfully capture the dynamics of flow discharge and stable isotope in the same study catchment. Similarly, elsewhere, Chang et al. (2020)

developed a model capturing the functioning of a dual flow system (fast flow and slow flow), showing that setting a passive storage in the fast flow reservoir can reproduce the dynamics of flow discharge and spring EC.

Our study was novel in comprehensively analyzing the functioning of alternative configurations of passive storage in a complex model structure for cockpit karst

catchments, based on a comparison of the performances of 14 different models. We demonstrated through this comparison that adding passive storage in the fast flow reservoir in hillslope unit is more efficient for simulating flow components and isotope dynamics, with three alternative choices to set passive storages in our developed model. The most parsimonious model is to add a passive storage in the hillslope fast

605 flow reservoir, as with model *c*. The "best" model is to add two passive storages in

fast flow and slow reservoirs of the hillslope unit, as is the case with model *f*. This best

model can appropriately estimate flow components in addition to the total discharge

and isotope concentration at the catchment outlet. Adding an additional passive

storage in the depression slow flow reservoir, such as model *j*, does not further

610 substantially increase the simulation accuracy even though the model obtains higher

$KGE_q$ and $KGE_c$ in Table 6.

**5.2 The dominant transport processes: advection, dispersion or molecular diffusion?**

Generally, the transport process is largely controlled by advection with the tracer

615 travelling with water, though molecular diffusion in the slow velocity (or immobile)

zone, and hydrodynamic dispersion in the fast velocity (or mobile) zone also contribute

(Karadimitriou et al., 2016; Schumer et al., 2003; Wang et al., 2020). In karst flow

systems, larger fracture and conduit media have permeability ranging across several

orders of magnitude higher than matrix flow in micropores. In cockpit karst catchments,

620 the hillslope unit has a higher flow velocity, but longer flow paths to the outlet. Tracers

input farthest from the stream at the hillslope unit will undergo more dispersion

(Kirchner et al., 2001). In our study catchment, the hillslope unit has a higher flow

velocity as the outflow coefficient of fast flow in the modeled hillslope unit is much

greater than that of the depression unit (Table 7) for the best performing models (*c*, *f*,

625 and *j*). Meanwhile, configuring passive storage in the hillslope fast flow alone is

sufficient to damp the δD variability effectively. This context, points to that

hydrodynamic dispersion dominates the chemical mixing. Indeed, the dominance of hydrodynamic dispersion has been widely reported in flow-conductive (preferential flow) zones (Roubinet et al., 2012). For example, Zhao et al. (2019, 2021) used a

transient storage model (TSM) to study the tailing of breakthrough curves (BTCs) of tracers in karst conduits, with experimental results suggesting that the dispersion coefficient was positively correlated with the flow velocity.

The mass exchange fluxes ($E_{GMF}$ and $E_{GMS}$ in Eqs. (11) and (12)) between active and passive storages are calculated in Table 10. The mass exchange flux of hillslope fast

flow is greater than that of slow flow, and over 10 times larger than that of slow flow in depression unit. This result also supports that hillslope unit has stronger dispersion effects. Therefore, only when the functioning of the advection and dispersion of the hillslope unit is incorporated in the models, the stronger variations of discharge and isotopes can be better captured simultaneously.

**Table 10.** The simulated $|E_{GM}|$ ($m^3 \times ‰$) of flow components in the hillslope-depression-outlet continuum for the 30 optimal solution sets from the selected representative models during the study period

| No. of Passive storage | Model | Hillslope | | | | Depression and catchment outlet | | | |
|---|---|---|---|---|---|---|---|---|---|
| | | Slow flow | | Fast flow | | Slow flow | | Fast flow | |
| | | Range | Mean | Range | Mean | Range | Mean | Range | Mean |
| 1 | $c$ | - | - | 0-42519 | 122 | - | - | - | - |
| 2 | $f$ | 0-13776 | 35 | 0-51603 | 120 | - | - | - | - |
| 3 | $j$ | 0-19816 | 42 | 0-46338 | 106 | 0-5773 | 10 | - | - |

Note: "-" represents not available.

**5.3 Uncertainties of adding passive storage in the tracer-aided hydrological modeling**

Our model uses a distribution curve of unsaturated storage capacity to describe the spatial heterogeneity of storage volumes, and fast flow and slow flow systems to conceptualise dual karst flow systems on a large scale (e.g. hillslope and depression units). Optimizing the number of storages balances the need to minimize model complexity and uncertainty, while still improving the simulation performance of both flow and tracers. Particularly for karst catchments, this optimization needs to be based on short-time-interval observation data, such as hourly data in our study catchment to capture the rapid hydrological response. Only such fine resolution data can capture the dramatic variability of the hydrograph and tracer dynamic, and thus can be used to successfully optimize the model structure. Nevertheless, the optimized passive storages and model structures are not unique, as the three better models with 1-3 passive storages performed similarly well in the study catchment, in terms of the catchment input-output responses.

These uncertainties imply that additional observations are needed to enhance our ability to constrain complex model structures and ranges of model parameters in karst catchments. These additional observations should include not only the catchment inputs - output responses, but also some key hydrological internal state components and their isotope concentrations, such as water fluxes and isotope transport in micropore, fracture and conduit media in karst catchment. Moreover, detailed observations of human activities are also important to reduce the modeling uncertainties. As shown in our study

catchment, the depression is occupied by agricultural land. Groundwater pumping for

agriculture use causes the sudden declines in streamflow and isotopic concentrations in

June as shown in Fig. 7, which makes that the model overestimates low flow.

Our study catchment at Chenqi is broadly representative of extensive regions of

headwater catchments in cockpit karst landscapes, and while the model parameters still

need to be calibrated for specific catchments, the model is generic and transferable to

other areas. The approach also has the potential to be used in upscaling to large

catchments, though the model would then need to incorporate river and channel routing

as these play an important role in regulating streamflow variations at larger scales.

**6 Conclusions**

In cockpit karst landscapes dominated by poljes and surrounding tower areas,

depression areas are interconnected with isolated towers scattered throughout the

terrain (Lyew et al., 2007). In this study, we developed and tested a coupled flow-tracer

model for simulating discharge and isotope signatures for cockpit karst landscapes

represented as a "hillslope-depression-outlet" continuum. We tested 14 simulation

cases with alternative model structures by varying the number and configuration of

passive storage in the fast/slow flow reservoirs of hillslope/depression units. The model

structures and parameters were optimized using a multi-objective optimization

algorithm to match the observed discharge and isotope dynamics in the Chenqi

catchment of southwest China.

We found that for complex models developed for cockpit karst catchments, capturing the main hydrological flow paths and organising passive storages in relation to these flow paths can efficiently improve model performance. In the Chenqi catchment, the

main hydrological pathways are hillslope flow and its connection with the catchment outlet. The models with 1-3 passive storages achieve similarly optimal results that are supported by the values of $KGE_q$, $KGE_c$ and $Abias_q$. All three models have a passive storage in the dominant flow domain (hillslope fast flow).

The optimal model structure is supported by the simulated discharge and tracer

dynamics. The hillslope fast flow system contributes about ~80% of the outlet discharge. The passive storages in the optimal models strengthen isotope mixing and thus constrain the $\delta D$ and discharge variability. Further comparison of the simulated results by the three optimal models with 1-3 passive storages, showed the "best" model structure is to incorporate two passive storages in the fast and slow flow reservoirs of the hillslope

unit. This best model can appropriately estimate flow components in addition to the total discharge and isotope fluctuations at the catchment outlet.

Characterizing the dynamics of flow paths and connections in complex geological settings karst landscapes is central to better understanding fluid flow and solute transport processes. This study provided evidence that the protection of hillslope

environments is significant for the prevention of natural hazards, such as droughts, floods and contamination in karst landscapes.

**Acknowledgment:** This research was supported by the National Natural Science Foundation of China (42030506, 41971028). We thank Natalie Orlowski, the two

reviewers (Catherine Bertrand and the anonymous reviewer), Thom Bogaard and the

editor for their constructive comments which significantly improved the manuscript.

**Data availability:** The discharge and isotope data that support the findings of this study can be shared after the ending of our project according to the project executive policy. Anyone who would like to use the data can contact the corresponding author.

**Code availability:** The code that support the findings of this study is available from the corresponding author upon reasonable request.

**Author contribution:** GL was responsible for writing the original draft, methodology, data curation, and visualization. XC conceptualized the project, reviewed and edited the manuscript, conducted the formal analysis, and acquired the funding. ZZ and LW

developed the methodology and curated the data. CS reviewed and edited the manuscript.

**Declaration of Competing Interest:** I declare that neither I nor my co-authors have any competing interest.

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
