# Peer review of "Effects of passive storage conceptualization on modelling hydrological function and isotope dynamics in the flow system of cockpit karst landscape"

_Hydrology and Earth System Sciences, 2021_

## Referee Comment (RC2)

This article raises the problem of how to improve the knowledge of the functioning of karst aquifers by combining field data and a numerical model that wants to consider all flows reflecting different modes of transfer.

This study relies on numerous oxygen-18 isotopic data to better constrain the different volumes of water present in karst systems.

This study thus proposes an interesting approach but remains very local and does not propose interesting perspectives to other contexts.

The figures are not of good quality and are often too small for the information to be used quickly.

The bibliography lacks recent references and sometimes is not appropriate to support an argument. The introduction really needs to be improved by referring to more recent and relevant work.

For example, citing the 2003 paper by Batiot et al. to refer to the fact that oxygen isotopes can provide information on water residence times is a misuse of this work since in this paper Batiot et al. use TOC and Mg as a tracer of fast transit times versus long residence times. There are no references to isotopes in this paper. Again, the citations should be reviewed as there is recent work on the use of isotopes to improve knowledge of karst systems.

In line 62, the authors refer to work from 2010 and 2013 as the state of the art of models at different scales of study that have been developed to describe flows in karst. There is recent work on tracing-modelling coupling by the Montpellier team that could have been used to support the authors' argument.

Finally, to end these comments on bibliographic references, the work of Rodriguez et al. (2017) is cited on line 127 but the reference does not appear in the bibliographic list.

On the background of the article

Introduction

In my opinion, the introduction is a bit confusing and would benefit from being reworked and clarified especially in the justification section of the study. The authors go directly from the general idea to the application on their site without explaining why their site will allow them to answer their problem if only because there are isotopic and hydrological data (which ones).

Page 88; can the authors clarify this concept "Hence, the storage...." How do they account for the seasonality of water isotopic levels and their notion of storage?

On the study site part

This paragraph should also be reworked, especially figure 1 which is unclear.

It is difficult to distinguish the sources on the figure.

I would have liked to have a more complete description of their karstic system. Where are located the two epikarst springs mentioned in line 168? Are they the two pink triangles?

Where is the main outlet of this system located, are there any isotopic and hydrological data? I asked myself this question while reading the description of the hydrological response of epikarst springs to

precipitation. It is difficult to say that the behaviour of epikarst springs reflects the behaviour of the karst system itself.

This raises the question of what the authors want to identify in their article, is it to work on flows in the epikarst or in the karst? In which case the problematic of the introduction must be reoriented and the bibliography better targeted.

In the "Obervationnal dataset" section, it would have been nice to structure this paragraph better between data collection and isotopes analysis

The first part of this paragraph concerns data acquisition

Were the samples collected in the automatic samplers analysed quickly to avoid evaporation problems?

Can you provide details on how the groundwater was collected?

Is it possible to have a little more detail on the dates of sampling? Which samples were taken at the same time, what is the time lag between rainwater and groundwater?

The second part of this paragraph concerns the analysis of isotopic data.

Figure 2 really needs to be taken back because it is unreadable. I can't follow their reasoning based on this figure.

What is the significance of some correlations that have coefficients at 0.21?

Where are the sources of the hillslope?

Line 216 "this phenomenon....recharge" is this really surprising? do we need so much isotopic analysis to reach this conclusion?  What do the authors want to demonstrate? Or rather, what do they bring that is new?

I think that this paragraph really needs to be reworked by providing information on the geometry of their system, to make figure 2 readable, and to explain the variability of the results of each analysis point. This figure brings more confusion than help in the argumentation.

It would also be necessary to specify the precautions of the mode of sampling especially for the analysis of isotopes. Finally, it would be necessary to have a temporal idea of the samples at each sampling site. This could help in the analysis of the results.

Finally, how can we consider a flow model, a tracer that is not conservative? Doesn't this call into question their initial hypothesis concerning the fact of using a tracer to identify stored water volumes

Model development part

I am well aware that one has to start from hypothesis to build a conceptual model that helps to lay the foundations of a numerical model, but I am not sure that considering the epikarst as an analog of a karst system is really relevant. A better justification than the one given is really needed. The calibration of the model with a tracer which is supposed to be conservative, and which is not, given the evaporation curves. Even if the results between calibration and validation are satisfactory, it is the very design of the model that is problematic.

Where do the hydrographs in figure 5 come from? This was not mentioned in the data section. Or how was it measured?

Is taking into account a certain number of passive storages until arriving at a satisfactory modelling result representative of reality?

The conclusion also needs to be reviewed and above all, what prospects are there for extending this study to other cases?

It would have been nice to analyze the relevance of the conceptual model (epikarst as an analog of a karst) to give some weight to their study and try to bring some opening elements.

It remains a very local study, with results that seem coherent, but on what assumptions?

---

## Author Comment (AC1)

**Title: Effects of passive storage on modelling hydrological function and isotope dynamics in a karst flow system in southwest China**
**Author(s): Guangxuan Li, Xi Chen, Zhicai Zhang, Lichun Wang, and Chris Soulsby**
**MS No.: hess-2021-492**
**MS type: Research article**

**Responses to the Reviewers:**
**Reviewer #1: This manuscript needs to below corrections:**
The paper presented by Li et al. deals with the internal organisation of hydrological systems in terms of the number of reservoirs involved, interactions between these reservoirs and their relative contributions. It is a classical conceptual approach comparable to that of global hydrological models but enriched here by the contribution of tracer data. The article is well written overall, well structured and the illustrations are of good quality (except for figures 6 and 7 which are difficult to read because of the chosen scales). The objectives are clearly stated and the methods used are appropriate and sound. This approach is not, however, original and is a contribution to the series of studies that have been carried out for several years on the contribution of isotopic data to improving the structure of hydrological models (see Uhlenbrook S, Leibundgut C. 1999 for one of the first studies in recent advances). The list of references appears well balanced at first glance with about one third of the references cited being less than 5 years old and half being less than 10 years old. There is little very recent literature on the understanding and modelling of karst systems or on coupled flow-isotope modelling involving questions on mixing processes, residence time distribution or the relationship between velocity and celerity. On the other hand, one third of the articles cited that are less than 5 years old already concern the basin studied (+ 2 other older articles), one of which mentions a coupled hydrology-isotope model. The topic is therefore promising, but we must ask ourselves how this new study improves our knowledge of the system and whether we have made any progress in terms of conceptualization. Apart in the introduction, this question is never addressed and as it stands it does not seem that a totally convincing conceptual scheme has been proposed. In particular, there is too great a disconnection with the field. Beyond the relative adequacy with the flow and isotope data, how does the structure of the model match the morphology of the catchment, underground and on the surface? (A broader discourse is also missing. The authors partly answer the initial question (line 111) but this only concerns the micro site studied. Can the proposed structure be generalised to larger areas (in comparable cockpit karst contexts) ?
Reply:
    We thank the reviewer for their valuable comments and suggestions. In the revision we will cite other more recent publications associated with the coupled flow-isotope modelling of karst system and associated contents.
    Cockpit karst landscapes are common in the tropics and sub-tropics area. The cockpit karst covers an area of about 140,000~160,000 km$^2$ in China. Such karst morphology

also exists in Southeast Asia, Central America and the Caribbean. Our selected catchment of Chenqi is a karst experimental catchment focused on investigations of hydrological, ecological and geological (carbonate dissolution) changes under climate change and human activities. So there are detailed observational data and field investigations in this catchment. The relevant publications cited in this study are necessary to provide the background context of our new model development and analysis. Although Chenqi catchment is small, the geomorphologic characteristics can represent a broad region of headwater catchments in cockpit karst landscapes.

In the polje/tower karst areas, the depression is more interconnected with isolated towers scattered throughout the terrain (Lyew et al., 2009). Geological surveys and observations show the hillslope unit lacks surface flow, and the depression unit has surface and underground drainage networks in such karst areas, including our study catchment. Understanding of interconnections of flow systems are vital for developing conceptual hydrological models for cockpit karst landscapes. Our new model presented in the paper is based on the coupled hydrology-isotope model developed by Zhang et al. (2019), a co-author of this manuscript. In this earlier model, the cockpit karst catchment was divided into two morphological units (hillslope and depression) and three water storage compartments (reservoirs) (hillslope reservoir, fast flow and slow flow reservoirs in depression). We substantially improved the model structure with a binary flow system (fast flow and slow flow ) in the hillslope unit, and the functioning of a binary moisture storage system of unsaturated zone (see Fig 4 in the original manuscript). Moreover, we optimized the model structure with a varying number of passive storages at different positions of the flow system (e.g. fast/slow flow reservoirs combined with different hillslope/depression units) based on a multi-objective optimization algorithm for best matching detailed observational data of hydrological processes and isotope concentration in the Chenqi catchment.

We agree there are various connections between hillslope and depression fast/slow flow reservoirs, and the model structure can be further improved in terms of the geomorphological surveys of the catchment. So, we set another reasonable connection between hillslope and depression fast flow and slow flow systems, and re-calibrated and validated the model (see descriptions below). We referenced the previous investigated results and will show more detailed geomorphological data in the revision (e.g. electrical resistivity tomography (ERT) image in Fig S1) to show how data has informed the evolution of this new model.

[Figure]

**Figure S1**. ERT image in the study depression. They interpret the ERT results as (a) an upper layer consisting of moist soils or extensively fractured rock (marked in blue); (b) carbonate rock with a high secondary porosity (and hence permeability; marked in light blue/yellow); (c) an underlying carbonate rock with low secondary porosity and hence relatively low permeability (marked in red) (Chen et al., 2018).

Additional reference:

Lyew-Ayee, P., Viles, H, A., Tucker, G, E.: The use of GIS-based digital morphometric techniques in the study of cockpit karst, Earth Surf. Process. Landforms., 32, 165-179, https://doi.org/10.1002/esp. 1399, 2009.

**(1) Lines 60-65: the list of references could be extended by some more recent articles (<10 years)**

**Reply:**

We carefully read the two references suggested by the reviewer, and added associated publications in the most recent 10 years as follows:

**The residence time:**

Brki, Z., Kuhta, M., Hunjak T.: Groundwater flow mechanism in the well-developed karst aquifer system in the western Croatia: Insights from spring discharge and water isotopes, CATENA., 161,14-26, https://doi.org/10.1016/j.catena.2017.10.011, 2018.

Zhang, Z., Chen, X., Cheng, Q., Soulsby, C.: Characterizing the variability of transit time distributions and young water fractions in karst catchments using flux tracking, Hydrol. Process., 34, 15, https://doi.org/10.1002/hyp.13829, 2020b.

**Modeling in karst:**

Husic, A., Fox, J., Adams, E., Ford, W., Agouridis, C., Currens, J., Backus, J.: Nitrate Pathways, processes, and timing in an agricultural karst system: Development and application of a numerical model, Water Resour. Res., 55, 2079-2103, https://doi.org/10.1029/2018wr02370, 2019.

Xu, C., Xu, X., Liu, M., Li, Z., Zhang, Y., Zhu, J., Wang, K., Chen, X., Zhang, Z., Peng, T.: An improved optimization scheme for representing hillslopes and depressions in karst hydrology, Water Resour. Res., 56, e2019WR026038, https://doi.org/10.1029/2019WR026038, 2020.

Ollivier, C., Mazzilli, N., Olioso, A., Chalikakis, K., Carrière, S.D., Danquigny, C., Emblanch, C.: Karst recharge-discharge semi distributed model to assess spatial variability of flows, Sci. Total Environ., 703, 134368, https://doi.org/10.1016/j.scitotenv.2019.134368, 2020.

**Hydraulics in karst:**

Ding, H., Zhang, X., Chu, X., Wu, Q.: Simulation of groundwater dynamic response to hydrological factors in karst aquifer system, J. Hydrol., 587, 124995, https://doi.org/10.1016/j.jhydrol.2020.124995, 2020.

**Mixing processes in karst:**

Dar, F., Jeelani, G., Perrin, J, Ahmed, S.: Groundwater recharge in semi-arid karst context using chloride and stable water Isotopes, Groundwater for Sustainable Development., 14, 100634, https://doi.org/10.1016/j.gsd.2021.100634, 2021.

Lorette, G., Viennet, D., Labat, D., Massei, N., Fournier, M., Sebilo, M., Grancon, P.: Mixing processes of autogenic and allogenic waters in a large karst aquifer on the edge of a sedimentary basin (Causses du Quercy, France), J. Hydrol., 593, 125859, https://doi.org/10.1016/j.jhydrol.2020.125859, 2021.

**(2) Lines 110-111: the question here is answered in the specific case of the study site. In what way is the structure of the model finally proposed transposable elsewhere and at a different scale? Are all cockpit systems of the same nature in terms of their hydrological functioning?**
**Reply:**
**Lines 110-111** "Particularly, the effects of passive storage structures are underexplored in terms of the location and number of passive storages needed for fast and/or slow flow reservoirs in hillslope and/or depression units, respectively. Consequently, it remains unclear what is the most efficient way of incorporating passive storage into coupled flow-tracer simulations."

In the original manuscript, we focused optimization of passive storages in hillslope and depression fast/flow reservoirs. We set fourteen schemes (scenarios) that incorporate 0~4 passive storages into different positions within the karst flow system, i.e., fast and/or slow flow reservoirs in combination with the hillslope and/or depression units (Table 3). We obtained the optimal structure (model $f$) for the coupled flow-isotope model that incorporated two passive storages in fast flow and slow flow paths of the hillslope unit.

This optimal structure is obtained based on the hydrological connections of hillslope - depression fast flow (HF-DF) and hillslope - depression slow flow (HS-DS). We further consider another possible connection of hillslope - depression fast flow (HF-DF), and hillslope slow flow (HS)- depression fast/slow flow (HS-DF/DS) with a ratio of $r_{hd}$ of HS contributing to DS (Fig. S2). The optimized $r_{hd}$ is 0.39. It means that about 61% of hillslope slow flow can enter depression fast flow reservoir. The optimal model structure of the passive-active storage connections is the same as the previous result (model $f$ in Fig. S2) while the optimized parameter values and hydrological components have some differences (See Table S2~S5).

[Figure]

**Figure S2**. Conceptualized structure for the coupled flow-isotope model for hillslope and depression unit connection. The light blue shades indicate active storage, the dark blue shades indicate passive storage.

From a geomorphological aspect, in the polje/tower karst areas, the depression is more interconnected with isolated towers scattered throughout the terrain (Trudgill, 1985). We proposed the concept of "hillslope-depression-stream" continuum that can capture the morphologic features of the cockpit systems (Chen et al., 2018; Zhang et al., 2020a). So, the developed model was based on the spatial discretization of hillslope and depression units, each with characteristics dominating runoff generation processes, streamflow processes and hydrological connectivity. In our study, the runoff generation is estimated based on water balance in unsaturated zone storage, the streamflow processes are routed by hillslope and depression fast and slow flow reservoirs, and hydrological connectivity includes connections of unsaturated zone (recharge)-

saturated zone (storage), and hillslope (fast and slow) flow - depression (fast and slow) flow. We believe that this model structure captures the internal catchment processes and hydrologic pathways of cockpit systems. Since hillslope runoff is regarded as a "water tower" for supplying the depression agriculture. Understanding the hillslope and depression hydrological functionality and their connections is necessary.

In our model, we used a distribution curve of the unsaturated storage capacity to describe the spatial heterogeneity of storage volumes, and fast flow and slow flow systems to elucidate dual karst flow system on a large scale (e.g. hillslope and depression units). Such delineations have been proven to be effective in other conceptual models, such as the VarKarst model (Hartmann et al., 2013) and the Xinanjiang model (Zhao, 1992). Surely, the model parameters still need to be calibrated when the model is applied to other catchments, but in principle the modeling approach is transferable. In large catchments, the model should incorporate river and channel routings that can play an important role in streamflow variations.

Additional reference:

Trudgill, S, T.: Limestone geomorphology. Longman, London, 196p, https://doi.org/10.1016/0012-8252(87)90065-1, 1985.

**(3) Study area section: The description of the site, especially the depression area, is very small. As expected, the average soil thickness is lower in the hillslope unit than in the depression unit. It is also expected that the nature of the soils is different and therefore also the field capacity. Can the authors provide details on these field characteristics? Also, please explain the phrase "perennially flowing underground conduit connecting the hillslopes to the catchment outlet" (see also line 337-338). Do you mean that there is a main karst conduit within the depression that transmits water to the outlet?**
**Reply:**

We have done lots of in situ sampling and analysis as well as field surveys by using electrical resistivity tomography (ERT) in the study catchment, particularly in the flat depression (see Fig S1). In depression, the accumulated soils are thick (~200 cm) and cultivated for crops of corn and rice paddy. The soils are most silt loam consisting of over 80% of clay and silt with soil particle size of smaller than 0.02mm and bulk density of 1.31 g/cm$^3$. The soil porosity ranges from 32% to 47%. In the hillslopes, Quaternary soils are thin (less than 30 cm) and irregularly developed on carbonate rocks. Outcrops of carbonate rocks cover 10%~30% of the hillslope area. The soil at sites from the shallow to deep layers in the catchment varies from sand loam, consisting of mostly sand (56~80%) and fine sand (20~40%), to calcareous soil and silt (1~10%). The bulk densities increased with depth, ranging from 1.02 to 1.33 g/cm$^3$ (Chen et al., 2009).

The depression unit has an underground channel/conduit system with perennial flow (see blue color in Fig S1). The high permeability zones (conduits) are sporadically distributed at the upper depression (the hillslope foot), which collects the hillslope flow. These connections can be identified by the slow recession of water table for the well

(W4) at the foot of hillslope after rainfall ceases (Chen et al., 2018). The widely distributed conduits in the upper depression are gradually concentrated to an underground channel at the catchment outlet.

Additional reference:

Chen, X., Zhang, Z., Chen, X., Shi, P.: The impact of land use and land cover changes on soil moisture and hydraulic conductivity along the karst hillslopes of southwest China, Environ, Earth Sci., 59, 811-820. https://doi.org/10.1007/s12665-009-0077-6, 2009.

**(4) What is there between the 2 m depth at the base of the soil and the water level reached at 13-30 m (see line 187)? How deep is the bedrock? What is its nature? What is the nature of the water table in the depression? It is doubtful that we are still in a karst system of the same nature as the hillslope.**
**Reply:**

The depression aquifer consists of the soil layer (about 2m thickness) overlying the lower fractured rocks according to the ERT image (see blue and light blue/yellow in Figure S1) and the drill core sampling (see Fig 3 in Chen et al., 2018). So, the depression aquifer has the bedrock (the impervious marlite formation) at depths of 30~50 m.

On line **187,** the depths of 13 ~ 35 m below ground surface refer to the sampling depth of groundwater, instead of the water levels.

The depression is located in the low-elevation area (<1340 m) and steeper hillslopes have high elevations ranging from 1340 to 1500 m as shown in Table 1. The water level ranges are 1,267.4~1,275.9 m at W1 with a mean of 1273 m, and 1,280.0~1,285.2 m at W4 with a mean of 1282 m.

**(5) Fig 1: there are 2 points for the outlet. Please define a single outlet for the catchment area. The location of the springs is not indicated on the map (only one spring? there is only one point on figure 1. Please specify)**
**Reply:**

There is a main underground channel in depression with an ascending spring at the catchment outlet, and high flows can spill over bottom of the depression ditches (referring to surface river channel with overland surface flow in Fig S2). So, in Fig 1, the two points at the outlet refer to the observation sites of underground and surface river channels at the catchment outlet (see Fig S3). The discharge used for simulations is the total of subsurface and surface discharge (see Fig S4). The discharge of surface flow and underground flow is 45% and 55% of the total discharge, respectively.

Two hillslope springs can be observed in the study catchment. We selected a perennial spring at the hillslope foot. The location has been added in the figure (see Fig S3).

[Figure]

**Figure S3**. The location of Chenqi catchment (a), surface river outlet (b), subsurface outlet (c) and hillslope spring(d).

[Figure]

**Figure S4**. The observed surface, subsurface and catchment total outlet flow (discharge)

**(6) Lines 164-166: Do the hydrographs mentioned refer to those observed at the outlet? There is ambiguity because the following sentence refers to epikarst springs.**
Reply:
   Lines 164-166: The hydrographs refer to the total discharge of surface and subsurface streamflow (see Fig S4).

The hillslope springs are formed by an impermeable layer (marlite) underlying the fracture zone (epikarst). We will change epikarst springs to hillslope spring to avoid misunderstanding.

*Observational dataset:*

**(7) The interpretation of figure 2 is very questionable, especially the regression line of the W1 points. There is a very large dispersion but the points W1, W4 and hillslope spring are to be included in the same O/D relationship which is also that of the local rainfall. The purple, green and black lines are disturbed by the few points indicating evaporation. The grey points (outlet) under the rainfall line are divided into two groups, probably indicating an evaporation process under different relative humidity conditions. In my opinion, it is not possible to argue about the age of the water from isotopic enrichment or depletion information alone:**

**1) The differences between the means for each set are modest and the number of measurement points is different each time. The difference in the mean between W1 and W4 is of the order of the measurement error. These differences are therefore not statistically interpretable.**

**2) Can't the apparent enrichment of W1 and outlet (vs W4) come from the inclusion of evaporated water? the apparent enrichment of W1 and outlet**

**3) Why is the dispersion on W1 the lowest? Could there be a different origin of water in W1 and W4? (linked to the organisation of the fracture network in hillslope).**

**4) Also specify the number of points and the origin of the data for the LWML.**
**Reply:**

We agree that the enriched groundwater is caused by evaporative isotopic fractionation. After checking the data points, we find that the two groups of the more enrichments of $\delta D$ and $\delta^{18}O$ with $\delta^{18}O>-7‰$ in Fig S5 occur in different periods for the outlet streamflow and hillslope spring. The upper points occur in the period of late spring ~early summer (May~ early June) with consecutive occurrence of small rainfall events while the lower points mostly occur in some summer days (middle July and August) after a long period without rain. In the period of late spring ~early summer, rain water is more enriched (see Fig. 7 in the original manuscript), resulting from the Westerly water vapor with low humidity and local moisture recycling, and thus the enriched rainfall infiltration controls isotopic concentrations of the hillslope spring and the outlet streamflow. By contrast, in the large rainfall period of summer, rainwater is depletion (see Fig. 7 in the original manuscript) due to the Western Pacific water vapor source with high humidity. Nevertheless, evaporation is strong in the dry period of summer. As streamflow recesses rapidly in the catchment, the low flow after a long drought period mostly comes from aquifer storage with strong evaporative isotopic fractionation.

The differences in $\delta D$ and $\delta^{18}O$ values at sites are related to the extent of mixture with new water (e.g., from rainfall recharge). For W1 close to the catchment outlet, the

site is located at a locally confined aquifer underlain by rocks with poor permeability according to the ERT survey. So, the subsurface flow seldom mixes with new water (rainfall) (Chen et al., 2018), resulting in more enrichment of groundwater at W1.

The plot of the δD and δ18O relationship is not directly related to water age. We will delete this description.

[Figure]

**Figure S5.** Plot of δ18O-δD for catchment outlet discharge and hillslope spring

(1) In Table 2, the sampling time is the same (wet season) for W1 and W4 although the sampling frequency is less than that of the hillslope spring and outlet discharge. The differences of the mean δD and δ18O values between W1 and W4 are larger than the measurement error (± 0.5 ‰ for δD and ± 0.1 ‰ for δ18O). To be comparable, we recalculated the statistical values using the data in the same period (Table S1). Although the mean and range of δD and δ18O values in Table S1 are different to those in the previous calculation in Table 2, they lead to the same conclusions as the previous result.

**Table S1**. Statistical characteristics of isotope data for rainfall, hillslope spring, catchment outlet discharge and depression groundwater in the period from the July 6, 2017 - August 20, 2017

| Obs | Numbers | δD (‰) | | | δ18O (‰) | | |
|---|---|---|---|---|---|---|---|
| | | Range | Mean | CV | Range | Mean | CV |
| Rainfall | 42 | -112.4~-32.7 | -77.5 | 0.26 | -14.7~-4.8 | -10.3 | 0.23 |
| Catchment outlet discharge | 255 | -73.1~-49.5 | -63.2 | 0.06 | -10.5~-5.5 | -8.8 | 0.08 |
| Hillslope spring | 252 | -77~-56.5 | -65.9 | 0.03 | -10.8~-6.1 | -9.4 | 0.06 |
| Groundwater W1 | 175 | -65.7~-50.7 | -60.8 | 0.03 | -9.6~-6.3 | -8.7 | 0.05 |

| | | | | | | | |
|---|---|---|---|---|---|---|---|
| Groundwater W4 | 47 | -70.2~-55 | -62.5 | 0.07 | -10.1~-7.9 | -8.9 | 0.07 |

(2) The depression groundwater at W1 and outlet flow is more enriched compared to that at W4, attributable to both evaporation and mixing with the new water (e.g. rainfall recharge). As W4 is located at the hillslope foot, and groundwater there receives more new water (fast flow) from hillslope; Table 2 shows the mean δD and δ18O values of W4 are closer to those of the hillslope spring and rainfall. W1 is located at a locally confined aquifer surrounded by rocks with poor permeability, and the flow seldom mixes with new water (rainfall) (Chen et al., 2018).

On the other hand, depression groundwater partly comes from rainfall infiltration and percolation through the thick soils (Zhang et al., 2019), which undergoes evaporative fractionation. Our re-optimized coefficient of the evaporative fractionation also supports this conclusion. As shown in Table S3 (model $f$), the coefficient of evaporative fractionation in depression ($ls_d = 0.05$) is greater than that of hillslope ($ls_h = 0.01$).

(3) The smallest coefficient of variation (CVs) is resulted from the less seasonal fluctuation of water table at W1 due to little rainfall recharge from the upper low permeability layer (refer to Figs 4 and 5 of Chen et al. (2018)).

(4) The LMWL of δD =8.18δ18O+9.52 comes from the daily rainfall sampled over the whole study period at Chenqi catchment. The number of points (253) for the LWML data is listed in Table 2. We have added a plot of the δD ~ δ18O data and the fitted line in Figure 2 (see Fig S6).

[Figure]

**Figure S6**. Plot of δ18O-δD for rainwater, catchment outlet discharge, hillslope spring and depression groundwater at wells W1 and W4

*Conceptual model structure :*

**(8) In connection with the study area section, one level of explanation is again missing for a satisfactory understanding of the system. The structure of the model logically foresees a dual flow system in the 2 units and in the 2 compartments ZNS-ZS. But it seems to me that the authors make two very strong assumptions that need to be justified:**

**1) The fast and slow reservoirs are in perfect connection between the unsaturated zone and the saturated zone. This can be understood in the karstic part of the hillslope system but the continuity does not seem so obvious in the depression part where the nature of the slow flow/fast flow partition can be quite different between the soil and the water table.**

**Reply:**

As shown in Fig 4 (and Fig S2), over most of the catchment area (i.e., the low permeability area of $\alpha$), runoff generated (free water R) in the unsaturated zone connects with both slow flow ($ks$) and fast flow ($1$-$ks$) reservoirs of the saturated zone, in which $ks$ is a discount coefficient of R entry into the slow reservoir (Fig S2). In the remaining area comprising the high permeability area ($1$-$\alpha$), rainfall directly enters underground channels through surface-connected sinkholes commonly found in carbonate aquifers.

In the depression unit, there are still some sinkholes that can accommodate rainwater even though the coverage ratio is small (e.g., 1% according to the re-optimized parameter of ($1$-$\alpha_d$) for model $f$ in Table S3).

**(9) 2) The authors suppose a hydrological continuity between the slow and fast flowing reservoirs of the 2 units (see also lines 330-331). Are there any tangible arguments to assume that slow flows from hillslope will retain this slow flow property in the depression (same for fast flows)?**

**Reply:**

Various lines of evidence have demonstrated the hillslope-depression fast flow connection. For heavy rainfall events, the observed hydrographs are primarily dominated by fast flow. In the mid-season after an extremely heavy rainfall, hillslope flow is highly synchronized with outlet flow, and the relationship between hillslope spring discharge and outlet discharge approaches a monotonic function (R.R. Zhang et al., 2020a). Figure S5 also shows that data of $\delta^{18}O$-$\delta D$ for the outlet discharge are strongly overlapping with those of hillslope spring for large rainfall events.

We agree that the hydrological connections of hillslope slow flow (HS) - depression fast/slow flow (DF/DS) are not perfectly conceptualised in our previous model structure (see Fig 4 in the original manuscript). When the depression water level is low and the storage deficit is high, a portion of HS could be concentrated into the depression conduits (DF). So, we redesigned connections of the flow system of the two units as shown in Fig S2 (i.e., HF - DF and HS-DF/DS connections). In this flow system, a parameter $r_{hd}$ is used to represent allocations of HS between DF and DS (Fig S2 and

Table S2). The optimized value of $r_{hd}$ is 0.39 for model $f$ (Table S3). Other associated variables are correspondingly recalculated, as shown in Table S4.

**Table S2.** The definitions of model parameters with their ranges

| Zone | Parameter and meaning | | Range |
|---|---|---|---|
| Area | $\alpha_h/\alpha_d$ | Ratio of matrix flow area | 0.90~0.95/0.95~1 |
| Unsaturated | $kc_h/kc_d$ | Coefficient for evapotranspiration | 0.9~1.3 |
| | $ks_h/ks_d$ | Ratio of water yield into slow flow reservoir | 0.1~0.5 |
| | $b_h/b_d$ | Exponential distribution of tension water capacity | 0.1~0.3 |
| | $ls_h/ls_d$ | coefficient of evaporation fractionation | 0~0.1 |
| | $wm_h/wm_d$ | Tension water storage capacity (mm) | 40~60/70~90 |
| | #$W_{h,pas}/W_{d,pas}$ | passive storage (mm) | 500~550 |
| Saturated | $Vm$ | Maximum storage of fast flow reservoir (mm) | 30~50 |
| | $r_{hd}$ | Ratio of hillslope slow flow into slow flow reservoir in depression | 0.1~0.8 |
| | $\eta s_h/\eta s_d$ | Outflow coefficient of slow flow reservoir | 0.001~0.01 |
| | $\eta f_h/\eta f_d$ | Outflow coefficient of fast flow reservoir | 0.01~0.15 |
| | $ke_h/ke_d$ | Exchange coefficient between slow and fast flow reservoirs ($10^{-3}$) | 0.1~1 |
| | #$\varphi_{sh}/\varphi_{sd}$ | Exchange coefficient between active and passive storages for slow flow | 0.1~0.5 |
| | #$\varphi_{fh}/\varphi_{fd}$ | Exchange coefficient between active and passive storages for fast flow | |
| | #$V_{sh,pas}/V_{sd,pas}$ | Passive storage for slow flow (mm) | 300~350 |
| | #$f_{fh,pas}/V_{fd,pas}$ | Passive storage for fast flow (mm) | |

Note: the upper and lower parameters and values in */* represent those in hillslope and depression, respectively; the parameters indicated by # refer to those used for isotope concentration simulation. - represents not available.

**Table S3.** The average calibrated values of model parameters for the 30 optimal solution sets from the selected representative models (model $a$, $c$, $f$, $j$, $n$)

| Zone | Parameter | Model $a$ | Model $c$ | Model $f$ | Model $j$ | Model $n$ |
|---|---|---|---|---|---|---|
| Area | $\alpha_h/\alpha_d$ | 0.95/0.98 | 0.92/0.99 | 0.94/0.99 | 0.94/0.98 | 0.93/0.98 |
| Unsaturated | $kc_h/kc_d$ | 1.13/1.19 | 1.14/1.08 | 1.12/1.04 | 1.17/1.15 | 1.19/1.15 |
| | $ks_h/ks_d$ | 0.33/0.27 | 0.24/0.13 | 0.22/0.14 | 0.16/0.23 | 0.3/0.19 |
| | $b_h/b_d$ | 0.21/0.18 | 0.14/0.24 | 0.11/0.15 | 0.24/0.15 | 0.11/0.17 |
| | $ls_h/ls_d$ | 0.04/0.05 | 0.01/0.01 | 0.01/0.05 | 0.02/0.02 | 0.02/0.05 |
| | $wm_h/wm_d$ | 50/85 | 58/90 | 56/82 | 52/81 | 50/77 |
| | #$W_{h,pas}/W_{d,pas}$ | 526/519 | 547/534 | 535/509 | 528/517 | 532/528 |
| Saturated | $Vm$ | 38 | 44 | 36 | 35 | 40 |
| | $r_{hd}$ | 0.65 | 0.37 | 0.39 | 0.55 | 0.51 |

| | | | | | |
|---|---|---|---|---|---|
| $\eta s_h/\eta s_d$ | 0.001/0.002 | 0.002/0.005 | 0.004/0.003 | 0.003/0.004 | 0.006/0.003 |
| $\eta f_h/\eta f_d$ | 0.09/0.02 | 0.15/0.01 | 0.14/0.01 | 0.14/0.02 | 0.14/0.11 |
| $ke_h/ke_d$ | 0.2/0.3 | 0.2/0.3 | 0.2/0.3 | 0.3/0.5 | 0.4/0.7 |
| #$\varphi_{sh}/\varphi_{sd}$ | -/- | -/- | 0.18/- | 0.22/0.29 | 0.23/0.22 |
| #$\varphi_{fh}/\varphi_{fd}$ | -/- | 0.25/- | 0.26/- | 0.19/- | 0.11/0.13 |
| #$V_{sh,\,pas}/V_{sd,\,pas}$ | -/- | -/- | 316/- | 331/323 | 346/320 |
| #$f_{fh,\,pas}/V_{fd,\,pas}$ | -/- | 322/- | 325/- | 334/- | 325/312 |

Note: the upper and lower parameters and values in */* represent those in hillslope and depression, respectively; the parameters indicated by # refer to those used for isotope concentration simulation. - represents not available.

**Table S4.** The proportions of flow components in the hillslope-depression-outlet continuum for the 30 optimal solution sets of the selected representative models during the calibration period (%)

| No. of Passive storage | Model | Hillslope | | | | Depression and catchment outlet | | | | | |
|---|---|---|---|---|---|---|---|---|---|---|---|
| | | Slow flow | | Fast flow | | Slow flow | | Fast flow | | Surface flow | |
| | | Range | Mean | Range | Mean | Range | Mean | Range | Mean | Range | Mean |
| 0 | $a$ | 5~38 | 24 | 62~95 | 76 | 5~23 | 14 | 38~78 | 55 | 0~56 | 31 |
| 1 | $c$ | 8~31 | 16 | 69~92 | 84 | 4~19 | 9 | 41~61 | 54 | 25~55 | 37 |
| 2 | $f$ | 7~32 | 15 | 68~93 | 85 | 4~19 | 9 | 39~58 | 49 | 28~57 | 42 |
| 3 | $j$ | 7~34 | 15 | 66~93 | 85 | 4~20 | 10 | 40~71 | 51 | 23~54 | 39 |
| 4 | $n$ | 8~31 | 19 | 69~92 | 81 | 4~18 | 10 | 37~89 | 74 | 2~56 | 16 |

Note: the contrasting pairs of models ($c$ vs. $a$, and $n$ vs. $j$) reflect effect of fast flow reservoir with an additional passive storage on flow components; the contrasting pairs of models ($j$ vs. $f$) reflect effects of slow flow reservoir with an additional passive storage on flow components.

**(10) Lines 235-241: these sentences are written as if to compare the nature of the slow and fast flows in the 2 units (hillslope vs depression epikarst vs upper soil). In this context, the sentence referring to fast flows speaks of large fractures vs. swallow holes, which suggests that the latter formations are in the depression part. Can you confirm this impression? If so, should this be linked to the "perennially flowing underground conduit" mentioned in the study area section and the "underground channel in depression" in line 337-338? Overall, the authors should make an effort to describe the hydrogeomorphological context of the system and better relate this information to the structure of the proposed model.**
**Reply:**

Your descriptions are correct. We will revise this part of the description and strengthen the geomorphological conceptualisation in the study catchment as described above.

**(11) 3.1.2 isotopic concentration routing: you do not take into account isotopic fractionation, whereas Figure 2 shows that there is evaporation. This fractionation is however integrated in the model proposed in Zhang et al (2019). Can you explain why you chose to ignore this process?**
**Reply:**

Based on your suggestion and the model proposed by Zhang et al. (2019), our we will developed out model further and include isotopic fractionation by adding an isotopic fractionation coefficient *ls*. Then the mass balance in the unsaturated zone storage can be expressed as:

$$\frac{d(WU\delta_b)}{dt} = P\delta_p - R\delta_b - E(1+ls)\delta_b \tag{S1}$$

where *WU* ($WU = W + W_{pas}$) is the moisture storage consisting of active storage *W* and passive storage $W_{pas}$, $\delta_p$ and $\delta_b$ are the stable isotope concentrations of rainwater (*P*) and moisture (and water yield *R*), respectively, and is the coefficient of evaporative fractionation.

As expected, the optimized *l*s value is larger in the depression unit ($ls_d = 0.05$) than in the hillslope unit ($ls_h = 0.01$) (see model *f* for Table S3).

**(12) 3.2 Model calibration and validation: this section is again too far from the reality of the field. The choice of parameter values to be set and calibrated must depend on the characteristics expected in the hillslope unit and in the depression unit. It is not obvious a priori to admit that the proportion of the matrix volume is the same in the two units. Similarly, the parameter *b* should be dependent on the nature of the matrix and that of the fast flow paths. Are these natures the same in the two units? Finally, how do you justify the same value of Wpas for both units and the fact that Wpas=Vpas?**
**Reply:**

The parameter ranges are set relying on the physical characteristics established in previous studies in this catchment. Some parameter values are directly specified according to field investigations (e.g., the ratio of matrix flow area $\alpha$) and suggestions by other studies (e.g., the storage capacity of moisture *wm* and the passive storage for slow flow and fast flow $V_{pas}$) (Xue et al., 2019; He et al., 2019; Liu et al., 2020).

We did further calibration for all parameters. The recalibrated parameter values are different to those of previous calibrations, but the parameter values in the hillslope and depression units are ordered similarly to the previous results. For example, the ratio of matrix flow area of $\alpha$ in hillslope is 0.94, smaller than that in depression (0.99); *wm* representing the soil moisture retention capacity of 56 mm for thin soils over hillslope, is much smaller than 82 mm for thick soils over depression (see the parameter values for model *f* in Table S3). *b* represents spatial heterogeneity of water storage capacity for the matrix of unsaturated zone, instead of the conduits (fast flow) because the conduit area has been separated from the area for each of the hillslope and depression units (see Fig S2).

The curves of storage capacity *WM* related to the proportion (f/F) of the matrix area for unsaturated zone are shown in Fig S7. It shows over half of the area with WM less than 62.2mm and 94.3mm for thin soil in the hillslope unit and thick soil in the depression unit, respectively.

[Figure]

**Figure S7**. Storage capacity curve

Most studies show that the volumes of passive storages ($W_{h,pas}$, $W_{d,pas}$, $V_{s,pas}$ and $V_{f,pas}$ in Table S2) are generally one order of magnitude larger than those of active storage (Dunn et al., 2010, Soulsby et al., 2011, Ala-Aho et al, 2017). When the parameters are calibrated in this study, the calibrated values are 535 and 509mm for the unsaturated zone passive storages ($W_{h,pas}$ and $W_{d,pas}$) in the hillslope and depression units, respectively, and 316 and 325mm for the saturated zone passive storages ($V_{fh,pas}$ and $V_{fd,pas}$) in the hillslope and depression units, respectively (see model *f* in Table S3). These large passive storages ensure damping and time-lags of $\delta^{18}O$ and $\delta D$ in streamflow response compared with precipitation fluxes, implying large mixing volumes that are usually much greater than dynamic storage changes estimated by water balance calculations (Birkel et al., 2011; Fenicia et al., 2010; Soulsby et al., 2011).

Additional reference:

Liu, Y., Zhang, K., Li, Z., Liu, Z., Wang, J., Huang, P.: A Hybrid Runoff Generation Modelling Framework Based on Spatial Combination of Three Runoff Generation Schemes for Semi-humid and Semi-arid Watersheds, J. Hydrol., 590, 125440, https://doi.org/10.1016/j.jhydrol.2020.125440, 2020.

**(13) Table 4: For the calibrated parameters, which model is presented on the 14 scenarios? I may have missed something but I don't understand the NA for φsd, φfd and for Vth-pas and Vfd-pas.**
**Reply:**

In the original manuscript, model *n* includes all the parameters associated with passive storage for calibration. Other models include different choices of passive storages for fast and slow flow reservoirs in the hillslope and depression units. For

example, model $f$ includes two passive storages of hillslope fast and slow reservoirs as the optimal model structure excludes other two passive storages of depression fast and slow reservoirs. So, the parameters of exchange coefficients between active and passive storage for the depression unit ($\varphi_{sd}$ and $\varphi_{fd}$ for model $f$ in Table 4) are not necessary. "NA" in Table 4 means not applicable. We will use "-" to replace "NA" in order to avoid misunderstanding.

**(14) Line 383: prefer μ to σ to express an average (or a ratio of averages)**
**Reply:**

We will change the expressions of the two variables.

**(15) Line 397: I am not very familiar with multi-objective optimisation algorithms but the number of iterations seems low. In fenicia et al (2007), the number of iterations is rather in the order of a few thousand.**
**Reply:**

We have executed the multi-objective optimisation algorithm again by increasing the number of iterations from 100 to 1000. The calibration of the NSGA-II algorithm was performed as follows:

For a number of iterations (e.g. 1000 in this study), 50 parameter sets were initially retained for screening out the sets with $Bias_q$ less than or equal to 0.2. Then the remaining sets are sorted from the largest to the smallest according to the sum of corresponding $KGE_q$ and $KGE_c$. Finally, 30 sets are selected as the Pareto-optimal solution (Nan et al., 2021). The corresponding objective function values (average of the optimal solution sets) for both the calibration and validation periods were extracted.

However, even with the number of iterations increased from 100 to 1000, we find that the results are not significantly different.

*4.1. Performance of models:*
**(16) Lines 428-429 (and Figures 6-7): The authors should be more critical about their results. In particular, the model does not really succeed in capturing isotopic variations. In many cases, it overestimates or underestimates the observed values, especially in the calibration period. In validation, the results are better because there is less variability. Finally, the performances are not better (or even worse) than those obtained with the model of Zhang et al. How can these shortcomings be explained in terms of the structure (defaults) of the model? The announced contributions of the fast flow reservoir are not inconsistent, but it is not reasonable to justify these results by those of another model. Once again, there is a lack of arguments from the field.**
**Reply:**

For the coupled modeling of hydrological and chemical or isotopic processes, all models give a higher accuracy of streamflow simulation than chemical or isotopic

simulations (Soulsby et al., 2015; Dehaspe et al., 2018; Mudarra et al., 2019; Birkel et al., 2020). Additionally, most isotopic simulations of the coupled models are compared with daily or monthly analysis data of the isotopic concentrations, and seldom compared with exacting test of hourly data as executed in our study. Our simulated accuracy is acceptable for the isotopic process with the $KGE_c$ of 0.59 and 0.73 in the calibration and validation periods, respectively, compared with that from other studies (Delavau et al., 2017; Aaron et al., 2019). The $KGE_c$ is also greater than 0.5 from the simulation of Zhang et al. (2019) in wet season.

The optimized model (model $f$) captures the sharp rise and decline of high flow and isotopic variations, but it can not simulate some low flow and isotopic processes similarly well. Particularly, the model overestimates some fast decreases of low flow caused by groundwater pumping for agriculture use (i.e., the sudden declines of streamflow and isotopic concentrations in June as shown in Figs S8 and S9). Comparatively, the validation period has fewer low flows and less groundwater pumping influences. As a result, the simulations are better in the calibration period than in the validation period.

We agree there are some limitations in our developed conceptual model due to the strong heterogeneity of the karst media. For example, some runoff can occur from areas of impermeable rocks despite small rainfall. We will discuss the limitations of the model in more detail in the revised manuscript.

We will strengthen descriptions of the field investigations used for comparison with our simulated results. As listed in Table S4, the simulation results of model $f$ suggested that the proportion of the total subsurface flow (slow flow and fast flow at underground channel) is 58%, and surface flow from the surface channel is 42% of the total catchment flow. These proportions are consistent with 55% and 45% from the observations at underground conduit outlet and surface channel outlet, respectively.

[Figure]

**Figure S8**. Simulated discharge concentrations of the 30 sets of optimal solutions by model *f* in calibration and validation periods.

[Figure]

**Figure S9**. Simulated isotope concentrations of the 30 sets of optimal solutions by model *f* in calibration and validation periods.

Additional reference:

Birkel, C., Duvert, C., Correa, A., Munksgaard, N. C., Maher, D. T., &Hutley, L. B.: Tracer-aided modeling in the low-relief, wet-dry tropics suggests water ages and

DOC export are driven by seasonal wetlands and deep groundwater, Water Resour. Res., 55, e2019WR026175, https://doi.org/10.1029/2019WR0261, 2020.

Delavau, C. J., Stadnyk, T., Holmes, T.: Examining the impacts of precipitation isotope input ($\delta_{18}O_{ppt}$) on distributed, tracer-aided hydrological modelling, Hydrol. Earth Syst. Sci., 21, 2595-2614, https://doi.org/10.5194/hess-21-2595-2017, 2017.

Dehaspe, J., Birkel, C., Tetzlaff, D., Sánchez-Murillo, R., DA, María., Soulsby, C.: Spatially distributed tracer-aided modelling to explore water and isotope transport, storage and mixing in a pristine, humid tropical catchment, Hydrol. Process., 570, 3206-3224, https://doi.org/10.1002/hyp.13258, 2018.

Mudarra, M., Hartmann, A., Andreo, B.: Combining experimental methods and modeling to quantify the complex recharge behavior of karst aquifers, Water Resour. Res., 55, 1384-1404, https://doi.org/10.1029/2017WR021819, 2019.

Neill, A, J., D, Tetzlaff., Strachan, N., Soulsby, C.:To what extent does hydrological connectivity control dynamics of faecal indicator organisms in streams? Initial hypothesis testing using a tracer-aided model, Hydrol. Process., 570, 423-425, https://doi.org/10.5194/hess-21-2595-2017, 2017.

Soulsby, C., Birkel, C., Geris, J., Dick, J., Tunaley, C., and Tetzlaff, D.: Stream water age distributions controlled by storage dynamics and nonlinear hydrologic connectivity: Modeling with high-resolution isotope data, Water Resour. Res., 51, 7759-7776, https://doi.org/10.1002/2015WR017888, 2015.

**(17) Table 5: How can it be explained that the parameters for the validation are better than those for the calibration? Did the authors consider switching the 2 periods?**
**Reply:**

Please refer to the above explanations why the simulated discharge and isotopic variations in the validation period are better than those in the calibration period. Since the calibration period is long and includes different magnitudes and variations of hydrographs, the calibrated parameters using the data are more representative than those from the data in the calibration period.

On the other hand, not all models obtain a higher accuracy of simulations in the calibration period. For example, models *d* and *h* in Table S5 obtain a lower simulation accuracy in the calibration period, compared to that in the validation period. These two models neglect passive storage of hillslope fast flow reservoir while they include passive storage of hillslope/depression slow flow reservoirs. No passive storage in the hillslope fast flow reservoir reflected the input signal overestimating stream isotopes. This was also reported by Birkel et al. (2011) for mountainous catchments in Scotland.

**Table S5.** The model performance based on the average of 30 optimal solution sets for individual model structure

| No. of Passive Storage | Model | Calibration | | | Validation | | |
|---|---|---|---|---|---|---|---|
| | | $KGE_q$ | $KGE_c$ | $Abias_q$ | $KGE_q$ | $KGE_c$ | $Abias_q$ |
| 0 | *a* | 0.46 | 0.30 | 0.08 | 0.46 | 0.38 | 0.23 |

| | | | | | | | |
|---|---|---|---|---|---|---|---|
| | b | 0.54 | 0.24 | 0.07 | 0.52 | 0.51 | 0.22 |
| 1 | **c** | **0.65** | **0.61** | **0.08** | **0.68** | **0.73** | **0.16** |
| | d | 0.42 | 0.31 | 0.09 | 0.4 | 0.04 | 0.25 |
| | e | 0.52 | 0.45 | 0.09 | 0.53 | 0.22 | 0.18 |
| | **f** | **0.68** | **0.59** | **0.09** | **0.72** | **0.73** | **0.14** |
| 2 | g | 0.47 | 0.39 | 0.1 | 0.48 | -0.12 | 0.19 |
| | h | 0.52 | 0.32 | 0.08 | 0.5 | 0.29 | 0.23 |
| | i | 0.65 | 0.15 | 0.07 | 0.67 | 0.5 | 0.12 |
| | **j** | **0.66** | **0.55** | **0.09** | **0.65** | **0.76** | **0.17** |
| 3 | k | 0.66 | 0.24 | 0.1 | 0.68 | 0.59 | 0.16 |
| | l | 0.63 | 0.21 | 0.08 | 0.64 | 0.32 | 0.14 |
| | m | 0.52 | 0.42 | 0.08 | 0.53 | 0.11 | 0.19 |
| 4 | **n** | **0.62** | **0.27** | **0.1** | **0.61** | **0.29** | **0.16** |

*4.2 The effect of number of passive storage*

**(18) It seems difficult to me to isolate the number of passive storage from their positions in the model. The comparison of models with different numbers of passive storage combines very different situations that are not very compatible (multiple combinations between slow flow vs fast flow and hillslope vs depression). The authors could consider combining parts 4.2 and 4.3 while trying to be more concise (e.g. be more synthetic on the damping effect of passive storage on isotope simulations)**

**Reply:**

We will concisely describe this as follows:

H and D represent hillslope and depression units, respectively;

F and S represent fast flow and slow flow, respectively;

P represents passive storage.

Thus, the dual flow system (F and S) combining with the two units (H and D) is represented by HF and HS, and DF and DS, indicating hillslope fast flow and slow flow, and depression fast flow and slow flow, respectively.

The flow system consists of hillslope and depression fast flow connection (i.e.. HF-DF), and hillslope slow flow (HS) and depression fast/slow flow connections (HS-DS/DF).

The passive storage (P) added in each flow reservoir (HF, DF, HS and DS) is represented by the subscript P, i.e., $HF_P$, $DF_P$, $HS_P$ and $DS_P$.

So, model a ~n could be described by:

**Table 3**. Different model structures that incorporate passive storages into fast flow and/or slow flow reservoirs at hillslope and/or depression units

| No. of Passive Storage | Model | Flow system |
|---|---|---|
| 0 | a | HF-DF and HS- DS/DF |

| | | |
|---|---|---|
| 1 | b | HF-DF and HS$_P$- DS/DF |
| | c | HF$_p$-DF and HS- DS/DF |
| | d | HF-DF and HS- DS$_p$/DF |
| | e | HF-DF$_p$ and HS- DS/DF$_p$ |
| 2 | f | HF$_p$-DF and HS$_p$- DS/DF |
| | g | HF-DF$_p$ and HS- DS$_p$/DF$_p$ |
| | h | HF-DF and HS$_p$- DS$_p$/DF |
| | i | HF$_p$-DF$_p$ and HS- DS/DF$_p$ |
| 3 | j | HF$_p$-DF and HS$_p$- DS$_p$/DF |
| | k | HF$_p$-DF$_p$ and HS$_p$- DS/DF$_p$ |
| | l | HF$_p$-DF$_p$ and HS- DS$_p$/DF$_p$ |
| | m | HF-DF$_p$ and HS$_p$- DS$_p$/DF$_p$ |
| 4 | n | HF$_p$-DF$_p$ and HS$_p$- DS$_p$/DF$_p$ |

We will merge 4.2 with 4.3.

**(19) Line 493: "observed values at the underground channel": what is this about? How were these observed underground flux values obtained? And why were these results not presented in part 2.2?**
**Reply:**
   See Figure S3 for the observations. This will be described in 2.2.

*4.4. The dominant transport processes*
**(20) Line 619: do you mean "hillslope unit"?**
**Reply:**
   Yes, it refers to the hillslope unit. We will correct it.

**(21) Line 622-625: peclet number = vL/D where D is the dispersion coefficient (since molecular diffusion is negligible in the context presented). If Pe hillslope > Pe depression, this means that advection (not dispersion) processes are more important in hillslope unit than in depression unit. This is consistent with the idea we have of the truly karst part of the catchment. But it does not seem to be consistent with a large exchange flow between active and passive storage (EGM) which effectively leads to a larger dispersion effect. Please clarify this point.**
**Reply:**
   Here, *Pe* is used to indicate advection or dispersion of the hillslope and depression flow movements (relating to flow velocity). The exchange between active and passive storages could be related to dispersion effect. We will revise these descriptions.

**Technical corrections :**

**(22) The article is generally well written. As I am not a native English speaker, there may be some improper sentence structures, but I did not have any major difficulties in following the development of the ideas. The general structure also seems to me fine.**
**Lines 292, 293, 443 and table 8: the "multiply" sign can be confused with the "minus" sign.**
**Reply:**

We will improve the manuscript writing.

We change all "·" to "×" as follows:

Line 272 on page 16:  $E = kc \times Ep \times \dfrac{W}{wm}$

Line 279-282 on pages 16-17:  $I_s = ks \times R \times \alpha$ , $((1\text{-}ks) \times I_s)$, $I_f = P \times (1\text{-}\alpha) + R \times (1\text{-}ks) \times \alpha)$

Line 290 on page 17:  $EX = ke \times (V_s\text{-}V_f)$
Line 292-293 on page 17:  $Q_s = \eta s \times V_s$, and $Q_f = \eta f \times V_f$
Line 308 on page 18:  $EXM = ke \times (V_s\text{-}V_f) \times \delta_s$, $ke \times (V_s\text{-}V_f) \times \delta_f$
Line 310 on page 18:  $\varphi_s \times V_s \times (\delta_s - \delta_{s,\,pas})$
Line 316 on page 18:  $EGM_f\ (= \varphi_f \times V_f \times (\delta_f - \delta_{f,\,pas}))$
Line 630 on page 36:  $|EGM|\ (m^3 \times ‰)$

**(23) Lines 582-585: Not clear. Please review this sentence structure.**
**Reply:**

This sentence is modified as follows:

Clearly, the location of passive storage is an important component of model structure, while optimizing the number of storage can balance minimizing model complexity and improving simulation performance of both flow and tracers.

---

## Author Comment (AC2)

Title: Effects of passive storage on modelling hydrological function and isotope dynamics in a karst flow system in southwest China Author(s): Guangxuan Li, Xi Chen, Zhicai Zhang, Lichun Wang, and Chris Soulsby MS No.: hess-2021-492 MS type: Research article

**Responses to the Reviewers:**

**Reviewer #2:**

dear editor

I thought long and hard before refusing this article, but in the end my arguments are as follows:

the bibliography is not up to date, which is annoying to put the study in an international context.

It remains a very local study and the lack of broadening and conclusive perspectives makes this article inappropriate for your journal.

Finally, the article (some figures illegible, bibliographic references badly cited or missing in the list) suggests that this work was done hurriedly.

Finally, this article can be accepted as a second intention with major revisions depending on your analysis

**Reply:**

We thank the reviewer for his valuable comments and suggestions.

Although Chenqi catchment is small, the geomorphologic characteristics can represent a broad region of headwater catchments in cockpit karst landscapes in the tropics and sub-tropics areas. The cockpit karst covers an area of about 140,000~160,000 km2 in China. Such karst morphology also exists in Southeast Asia, Central America and the Caribbean (Huang et al., 2014).

One of the hydrological characteristics of the cockpit karst landscapes is the hillslope - depression flow connections (H-D). In the karst area, since the flow system can be conceptualized into the fast flow (F) and slow flow (S) reservoirs in each of the hillslope and depression units, the hydrological connections include hillslope fast flow - depression fast/slow flow (HF-DF/DS), and hillslope slow flow- depression fast/slow flow (HS-DF/DS). As hillslope and depression fast flow (HF-DF) primarily moves in the connected conduits of the karst catchment, we neglected the connection of HF-DS in this study. Consequently, we consider three possible connections of hillslope - depression fast flow (HF-DF), and hillslope slow flow (HS)- depression fast/slow flow (HS-DF/DS) with a ratio of  $r_{hd}$  of HS contributing to DS (Fig. S1). The optimized  $r_{hd}$  is 0.39. It means that about 61% of hillslope slow flow can enter depression fast flow reservoir. The optimal model structure of the passive-active storage connections is the same as the previous result (model f) while the optimized parameter values and hydrological components have some differences (see Table S2~S5 in replies to the

reviewer

**Figure S1**. Conceptualized structure for the coupled flow-isotope model for hillslope and depression unit connection. The light blue shades indicate active storage, the dark blue shades indicate passive storage.

In reply to the reviewer 1, we have updated the references. Additionally, we summarized the previous studies that account for passive storages in hydrological models using at least one isotopic tracer (Table S1). It shows that number and location of passive storages are dependent on the model structure and the divided geographical units. Generally, the number of passive storages increases with the divided storages and geographical units. Therefore, for the complex karst flow system in the cockpit karst landscapes, the previous model structures with one passive storage (Zhang et al., 2019; Chang et al., 2020) may be insufficient to simulate the function of chemical mixing between active and passive storages. The optimized results from our generalized model structure incorporating all possible passive storages can make up for the deficiency.

Table S1. Summary of the previous studies that account for passive storages in hydrological models using at least one isotopic tracer

|       |       | Number        |                              |        |          |            |
|-------|-------|---------------|------------------------------|--------|----------|------------|
| Scale | Model | of
passive | Location of passive storages | Tracer | Function | References |
|       |       | storages      |                              |        |          |            |

|                      | Models with              |            |                             |                   |          | Domes and Donall      |  |
|----------------------|--------------------------|------------|-----------------------------|-------------------|----------|-----------------------|--|
| 25 ha                | fast and slow            | 1          | One storage                 | $^{2}\mathrm{H}$  | А        |                       |  |
|                      | flow reservoirs          |            |                             |                   |          | 1990                  |  |
|                      | Chemical-                |            |                             |                   |          |                       |  |
| $2.5 \text{ km}^2$   | mixing                   | r          | Shallow and deep            | Chlorida          | A and P  | Page at $a1 - 2007$   |  |
| 5.5 KIII             | dynamic                  | 2          | storages                    | Chioride          | A allu D | Page et al., 2007     |  |
|                      | TOPMODEL                 |            |                             |                   |          |                       |  |
| 23.6                 | The multiple             | 2          | C = 11 = t = m = = = | 211               | ٨        | Sam an al 2007        |  |
| km 2      | bucket model             | 3          | Soll storage                | -H                | А        | Son er al., 2007      |  |
| 2.0.1.               | The                      | 2          | Upper and lower             | 211               | ٨        | E-mining at al. 2009  |  |
| 5.8 na               | SoftModel i   | Z          | hillslope storages          | п                 | А        | Femicia et al., 2008  |  |
|                      | Complete                 |            |                             |                   |          |                       |  |
| 2.0.1.               | mixing                   | 1          | 0                           | 211               | D        | E-mining et al. 2010  |  |
| 3.8 ha               | and partial-             | 1          | One storage                 | -Ή                | В        | Fenicia et al., 2010  |  |
|                      | mixing model             |            |                             |                   |          |                       |  |
|                      |                          |            | 2 for upper and low         |                   |          |                       |  |
| 0.2                  |                          |            | storages in upper           |                   |          |                       |  |
| 2.5 and              | Lunan-CIM                | 25         | catchment, and 3            | 211               | A        | Birkel et al., 2011a  |  |
| 122                  | (L-CIM)                  | 2~5        | for upper, low and          | -H                |          |                       |  |
| Km-                  |                          |            | deep storages in            |                   |          |                       |  |
|                      |                          |            | lower catchment             |                   |          |                       |  |
| 3.6 and              |                          |            | The total of                |                   |          |                       |  |
| 30.4                 | SAM dyn model | 1          | The total of                | $^{18}$ O         | С        | Birkel et al., 2011b  |  |
| km 2      |                          |            | catchment storages          |                   |          |                       |  |
| 740                  | The tracer               |            | Shallow and deep            | 2 LI   |          |                       |  |
| 149                  | aided model              | 4          | storages for uplands        | П,
allvalinity | А        | Capell et al., 2012   |  |
| KIII                 | alded model              |            | and lowlands                | arkannity         |          |                       |  |
| 118                  | The                      |            | Unsaturated zona            |                   |          |                       |  |
| 1.4, 0               | DYNAMIT                  | 2          | and slow flow               | Chlorida          | A and D  | Hrachowitz et al.,    |  |
| 1 m 2     | (DYNAmic                 | 2          |                             | Chionde           | A allu D | 2013                  |  |
| KIII                 | MIxing Tank)             |            | reservoir                   |                   |          |                       |  |
|                      | Tracer-aided             |            |                             |                   |          |                       |  |
|                      | hydrological             |            | Three store res             | 180               | B and C  |                       |  |
| $201^{2}$            | model for a              | 3          | (upper lower and            |                   |          | Dirizol et al. 2015   |  |
| JU KIII              | wet Scottish             | 5          | (upper, lower and           | 0                 |          | Dirker et al., 2015   |  |
|                      | upland                   |            | saturation areas)           |                   |          |                       |  |
|                      | catchment                |            |                             |                   |          |                       |  |
|                      | Hydrochemical            |            | Shallow and                 |                   |          |                       |  |
| 3.7 km 2  | model of                 | 2          | groundwater                 | Chloride          | A and B  | Benettin et al., 2015 |  |
|                      | Upper Hafren             |            | storage                     |                   |          |                       |  |
|                      | The landscape-           |            | Three storages              |                   |          |                       |  |
| $3.2  \mathrm{km}^2$ | hased dynamic            | 3          | (hillslope,                 | 2 н    | R        | Soulsby et al 2015    |  |
| 3.2 XIII             | model                    | с э | groundwater, and            | 11                | Б        | 500150y et al., 2015  |  |
|                      | model                    |            | saturation area)            |                   |          |                       |  |

| 3.2 km 2                                  | STARR
(Spatially
Distributed
Tracer-Aided
Rainfall-                                   | 2 | Soil and
groundwater
storage    | 2 H                        | A, B and
C | van et al., 2016     |
|------------------------------------------------------|---------------------------------------------------------------------------------------------------|---|---------------------------------------|---------------------------------------|---------------|----------------------|
| 3.2, 0.6
and 0.5
km 2               | Runoff model)
STARR
(Spatially
Distributed
Tracer-Aided
Rainfall-
Runoff model) | 1 | Soil storage                          | 18 O                       | A and B       | Ala-Aho et al., 2017 |
| 3.2 km 2                                  | STARR model
for the humid                                                                      | 2 | Soil and
groundwater               | 2 H                        | A and C       | Dehaspe et al., 2018 |
| 10.2 ha                                              | A conceptual
catchment
model                                                                | 2 | Shallow and
groundwater
storage | 18 O                       | A, B and
C | Rodriguez., 2018     |
| 1.25
km 2*                             | Tracer-aided
hydrological
model for karst                                                   | 1 | Hillslope storage                     | 2 H                        | A, B and
C | Zhang et al., 2019   |
| 7.8 km 2                                  | STARR
(Spatially
Distributed
Tracer-Aided
Rainfall-
Punoff model)                  | 2 | Soil and
groundwater
storage    | 2 H                        | A, B and
C | Piovano et al., 2019 |
| Spring*                                              | Lumped
Model for
karst                                                                      | 1 | Fast flow reservoir                   | EC                                    | A             | Chang et al., 2020   |
| 0.23,
0.5, 0.6,
3.2 and
7.8 km 2 | A spatially
distributed
tracer-aided
hydrological
model
(STARR)                    | 1 | Soil storage                          | 2 H and
18 O | A, B and
C | Piovano et al., 2020 |
| 1.44
km 2                              | The EcH 2 O-iso
Model                                                               | 1 | The extra
groundwater
storage   | 2 H and
18 O | A, B and
C | Yang et al., 2021    |

[revised manuscript text omitted]

(1) This article raises the problem of how to improve the knowledge of the functioning of karst aquifers by combining field data and a numerical model that wants to consider all flows reflecting different modes of transfer.

This study relies on numerous oxygen-18 isotopic data to better constrain the different volumes of water present in karst systems.

This study thus proposes an interesting approach but remains very local and does not propose interesting perspectives to other contexts. Reply: Please see the above explanations.

**(2) The figures are not of good quality and are often too small for the information to be used quickly.**

**Reply:**

We will deliver the improved figures with high quality and clear information in the revised manuscript.

**(3) The bibliography lacks recent references and sometimes is not appropriate to support an argument. The introduction really needs to be improved by referring to more recent and relevant work.**

**Reply:**

We will revise the introduction to focus on hydrological connections of hillslope depression fast/slow flow in cockpit karst landscapes, and functions of passive storages incorporated into the total storage, particularly in karst flow systems, as summarized in Table S1. We have added associated publications in the most recent 10 years as follows:

**The residence time:**

- Brki, Z., Kuhta, M., Hunjak T.: Groundwater flow mechanism in the well-developed karst aquifer system in the western Croatia: Insights from spring discharge and water isotopes, CATENA., 161,14-26, https://doi.org/10.1016/j.catena.2017.10.011, 2018.
- Zhang, Z., Chen, X., Cheng, Q., Soulsby, C.: Characterizing the variability of transit time distributions and young water fractions in karst catchments using flux tracking, Hydrol. Process., 34, 15, https://doi.org/10.1002/hyp.13829, 2020b.

**Modeling in karst:**

- Dubois, E., Doummar, J., Pistre, S., Larocque, M.: Calibration of a lumped karst system model and application to the Qachqouch karst spring (Lebanon) under climate change conditions, Hydrol. Earth Syst. Sci., 24, 4275-4290, https://doi.org/10.5194/hess-24-4275-2020, 2020.
- Husic, A., Fox, J., Adams, E., Ford, W., Agouridis, C., Currens, J., Backus, J.: Nitrate Pathways, processes, and timing in an agricultural karst system: Development and application of a numerical model, Water Resour. Res., 55, 2079-2103, https://doi.org/10.1029/2018wr02370, 2019.
- Xu, C., Xu, X., Liu, M., Li, Z., Zhang, Y., Zhu, J., Wang, K., Chen, X., Zhang, Z., Peng, T.: An improved optimization scheme for representing hillslopes and depressions in karst hydrology, Water Resour. Res., 56, e2019WR026038, https://doi.org/10.1029/2019WR026038, 2020.
- Ollivier, C., Mazzilli, N., Olioso, A., Chalikakis, K., Carrière, S.D., Danquigny, C., Emblanch, C.: Karst recharge-discharge semi distributed model to assess spatial variability of flows, Sci. Total Environ., 703, 134368, https://doi.org/10.1016/j.scitotenv.2019.134368, 2020.
- Wunsch, A., Liesch, T., Cinkus, G., Ravbar, N., Chen, Z., Mazzilli, N., Jourde, H., and Goldscheider, N.: Karst spring discharge modeling based on deep learning using

spatially distributed input data, Hydrol. Earth Syst. Sci., 26, 2405-2430, https://doi.org/10.5194/hess-26-2405-2022, 2022.

Jeannin, P.Y., Artigue, G., Butscher, C., Chang, Y., Charlier, J.B., Duran, L., Gill, L., Hartmann, A., Johannet, A., Jourde, H., Kavousi, A., Liesch, T., Liu, Y., Lüthi, M., Malard, A., Mazzilli, N., Pardo-Igúzquiza, E., Thi éry, D., Reimann, T., Schuler, P., W öhling, T., Wunsch, A.: Karst modelling challenge 1: Results of hydrological modelling, J. Hydrol. 600, 126508, https://doi.org/10.1016/j.jhydrol.2021.126508, 2021.

**Hydraulics in karst:**

- Ding, H., Zhang, X., Chu, X., Wu, Q.: Simulation of groundwater dynamic response to hydrological factors in karst aquifer system, J. Hydrol., 587, 124995, https://doi.org/10.1016/j.jhydrol.2020.124995, 2020.
- Huang, W., Deng, C.B., Day, M.J.: Differentiating tower karst (fenglin) and cockpit karst (fengcong) using DEM contour, slope, and centroid, Environ. Earth Sci., 72, 407-416, https://doi.org/10.1007/s12665-013-2961-3, 2014.
- Jourde, H., Massei, N., Mazzilli, N., Binet, S., Batiot-Guilhe, C., Labat, D., Steinmann, M., Bailly-Comte, V., Seidel, J. L., Arfib, B., Charlier, J. B., Guinot, V., Jardani, A., Fournier, M., Aliouache, M., Babic, M., Bertrand, C., Brunet, P., Boyer, J. F., Bricquet, J. P., Camboulive, T., Carrière, S. D., Celle- Jeanton, H., Chalikakis, K., Chen, N., Cholet, C., Clauzon, V., Soglio, L. D., Danquigny, C., Défargue, C., Denimal, S., Emblanch, C., Hernandez, F., Gillon, M., Gutierrez, A., Sanchez, L. H., Hery, M., Houillon, N., Johannet, A., Jouves, J., Jozja, N., Ladouche, B., Leonardi, V., Lorette, G., Loup, C., Marchand, P., de Montety, V., Muller, R., Ollivier, C., Sivelle, V., Lastennet, R., Lecoq, N., Mar échal, J. C., Perotin, L., Perrin, J., Petre, M. A., Peyraube, N., Pistre, S., Plagnes, V., Probst, A., Probst, J. L., Simler, R., Stefani, V., Valdes-Lao, D., Viseur, S., Wang, X.: SNO KARST: A French Network of Observatories for the Multidisciplinary Study of Critical Zone Processes in Karst Watersheds and Aquifers, Vadose Zone J., 17, 180094, https://doi.org/10.2136/vzj2018.04.0094, 2018.
- Zhang, R., Chen, X., Zhang, Z., Soulsby, C.: Using hysteretic behavior and hydrograph classification to identify hydrological function across the "hillslope-depressionstream" continuum in a karst catchment, Hydrol. Process., 34, 3464-3480, https://doi.org/10.1002/hyp.13793, 2020a.

**Mixing processes in karst:**

- Dar, F., Jeelani, G., Perrin, J, Ahmed, S.: Groundwater recharge in semi-arid karst context using chloride and stable water Isotopes, Groundwater Sustain. Dev., 14, 100634, https://doi.org/10.1016/j.gsd.2021.100634, 2021.
- Lorette, G., Viennet, D., Labat, D., Massei, N., Fournier, M., Sebilo, M., Grancon, P.: Mixing processes of autogenic and allogenic waters in a large karst aquifer on the edge of a sedimentary basin (Causses du Quercy, France), J. Hydrol., 593, 125859, https://doi.org/10.1016/j.jhydrol.2020.125859, 2021.
- Mayer-Anhalt, L., Birkel, C., Sánchez-Murillo, R., Schulz, S.: Tracer-aided modelling reveals quick runoff generation and young streamflow ages in a tropical rainforest catchment, Hydrol. Process., 36, e14508, https://doi.org/10.1002/hyp.14508, 2022.

(4) For example, citing the 2003 paper by Batiot et al. to refer to the fact that oxygen isotopes can provide information on water residence times is a misuse of this work since in this paper Batiot et al. use TOC and Mg as a tracer of fast transit times versus long residence times. There are no references to isotopes in this paper. Again, the citations should be reviewed as there is recent work on the use of isotopes to improve knowledge of karst systems.

**Reply:**

We will delete this reference (Batiot et al., 2003) and cite the latest references about the use of isotopes to improve knowledge of karst systems as listed above (e.g., Brki et al., 2018; Zhang et al., 2020b).

**(5) In line 62, the authors refer to work from 2010 and 2013 as the state of the art of models at different scales of study that have been developed to describe flows in karst. There is recent work on tracing-modelling coupling by the Montpellier team that could have been used to support the authors' argument. Reply:**

We have updated the references and added more recent works on hydrological modelling such as Jourde et al.(2018), Dubois et al.(2020), Jeannin et al.(2021), and Wunsch et al.(2022) from recent works by the Montpellier team.

**(6) Finally, to end these comments on bibliographic references, the work of Rodriguez et al. (2017) is cited on line 127 but the reference does not appear in the bibliographic list.**

**Reply:**

We will add this reference as shown below:

Rodriguez, N. B., McGuire, K. J., Klaus, J.: Time-varying storage-Water age relationships in a catchment with a Mediterranean climate, Water Resour. Res., 54, https://doi.org/10.1029/2017WR021964, 2018.

**(7)**

**On the background of the article Introduction**

In my opinion, the introduction is a bit confusing and would benefit from being reworked and clarified especially in the justification section of the study. The authors go directly from the general idea to the application on their site without explaining why their site will allow them to answer their problem if only because there are isotopic and hydrological data (which ones).

**Reply:**

We will revise the introduction. Our selected catchment of Chenqi is a karst experimental catchment focused on investigations of hydrological, ecological and geological (carbonate dissolution) changes under climate change and human activities. So there are detailed observational data and field investigations in this catchment. The flow discharge was observed at intervals of 15 min, and water was sampled for isotope analysis at intervals of daily (dry season) and hourly (wet season). As we know, there are seldom detailed observations of isotope signatures. The previous coupled models of hydrological and isotopic processes (listed in Table S1) are mostly calibrated and validated against daily and weekly isotope signatures. In karst catchment, as flow discharge and isotope concentration vary dramatically fast, the coarse resolution data used in this study offers an opportunity to optimize our new model structure, such as hydrological connections of hillslope-depression fast/slow flow, and the functioning of passive storages in the karst flow system.

**(8) Page 88; can the authors clarify this concept "Hence, the storage...." How do they account for the seasonality of water isotopic levels and their notion of storage? Reply:**

Here the storage volume refers to the total storage (active storage and passive storage) for the isotope mixing (see Fig. S1). Passive storage does not directly contribute to streamflow, but it participates in stable isotope simulation (Hrachowitz et al., 2013). As shown in Eqs.  $(7)\sim(9)$  in the original manuscript, the passive storage added in the total storage takes a function of the isotope mixing and transport between active storage and passive storage and thereby can reduce the seasonality of isotopic composition in stream water.

**(9) On the study site part**

**This paragraph should also be reworked, especially figure 1 which is unclear. It is difficult to distinguish the sources on the figure.**

**I would have liked to have a more complete description of their karstic system. Reply:**

We have redrawn Fig. 1 as shown below (Fig. S2). There is a main underground channel in the depression with an ascending spring at the catchment outlet, and high flows can spill over the bottom of the depression ditches (referring to the surface stream in Fig. S2). So, in Fig. S2, the two points at the outlet refer to the observation sites of underground channel and surface stream at the catchment outlet. The discharge used for simulations is the total of underground channel and surface stream discharge.

Two hillslope springs can be observed in the study catchment (see Fig. 1 in Zhang et al. (2013)). We selected a perennial spring at the hillslope foot in this study. The location has been added in the figure (see Fig. S2). Water samples at two depression wells (W1 and W4 in Fig. S2) are analyzed, and the isotope compositions of W1 and W4 in comparison to those at the hillslope spring and the outlet discharge are used to indicate flow connections between hillslope and depression units.

---

## Author Response (AR1)

Title: Effects of passive storage on modelling hydrological function and isotope dynamics in a karst flow system in southwest China Author(s): Guangxuan Li, Xi Chen, Zhicai Zhang, Lichun Wang, and Chris Soulsby MS No.: hess-2021-492 MS type: Research article

**Responses to the Editor:**

Thank you for your letter and for the reviewers' comments concerning our manuscript entitled "Effects of passive storage on modelling hydrological function and isotope dynamics in a karst flow system in southwest China". Those comments were all highly valuable and helpful for revising and improving our manuscript, and they were an important guide to our research. We have studied those comments carefully and applied corrections to our manuscript accordingly. We have revised our manuscript, and we assure the reviewers that their concerns have been addressed properly. The revised sentences and sections in the revised manuscript are highlighted in blue color. The major modifications are summarized as follows:

(1) According to the comments of Reviewer 1, we have cited other more recent publications associated with the coupled flow-isotope modelling of karst system and associated contents (<10 years). We further improved the model structure by adding the evaporation fractionation process and resetting the connectivity between the hillslope and depression. After discussing the sensitivity of the parameters, we recalibrated the parameters of the model by using NSGA-II algorithm. Finally, the effect of passive storages is analyzed according to the latest calibration and validation results.

(2) According to the comments of Reviewer 2, we further clarified our scientific problem by using the newly added summary table (Table 1) on the application of passive storages in the coupled flow-isotope models. We not only described the differences between hillslope and depression by discussing hydrogeological properties, but also explained the sampling process of isotope observation data in detail. Finally, we expanded our discussion to explore the transferability of our model and approach.

(3) According to the comments of the editor, we have updated all the figures. In the part of results and discussion, we tried our best to compare our simulation results with previous studies (Table 1 in the revised manuscript), rather than simply listing the references in the introduction section.

Other minor issues raised by the reviewers have all been addressed accordingly.

We hope that the revision has addressed all the concerns of the reviewers.

Thank you for your editorial work.

Sincerely,

Xi Chen On behalf of all co-authors

Xi Chen, Professor of Hydrology Institute of Surface-Earth System Science, Tianjin University, Tianjin 300072, China E-mail: xi\_chen@tju.edu.cn \_\_\_\_\_

The list of all relevant changes in the revised manuscript, corresponding to the comments of reviewers and editor

**Reviewer #1:**

**Main comments:** Please refer to the page 2 lines 35-41, page 4 lines 51-58, page 5 lines 84-97, pages 37-39 lines 562-610 (**Section 5.1**) and pages 41-42 lines 644-674 (**Section 5.3**).

**Q(1):** Please refer to page 4 lines 62-63 and page 5 line 67.

**Q(2):** Please refer to pages 19-21 lines 261-279 (Fig.5), page 24 lines 361-370 and pages 25-26 lines 384-388 (Table 4).

**Q(3):** Please refer to page 4 lines 53-58, pages 12-13 lines 152-173 (**Section 2.2**) and pages 14-15 lines 199-209.

**Q(4):** Please refer to pages 12-13 lines 155-173 and pages 13-14 lines 187-193.

**Q(5):** Please refer to page 11 lines 144-147 (Fig.1) and page 15 lines 210-212 (Fig.2). **Q(6):** Please refer to pages 14-15 lines 200-212.

**Q(7):** Please refer to page 18 lines 251-253 (Table 3), page 34 lines 520-525 (Table 7), pages 15-16 lines 214-230 and page 17 lines 243-246 (Fig. 3).

**Q(8):** Please refer to pages 19-20 lines 270-279, page 19 lines 260-269 and pages 34 lines 520-524 (Table 7).

**Q(9):** Please refer to pages 16-18 lines 231-249 (Figs. 3 and 4), pages 28-29 lines 445-448 (Table 5), page 34 lines 520-524 (Table 7) and page 36 lines 553-556 (Table 8).

**Q(10):** Please refer to page 19 lines 260-269 and pages 12-13 lines151-173 (Section 2.2).

**Q(11):** Please refer to page 22 lines 325-328, pages 28-29 lines 445-448 (Table 5) and page 34 lines 520-524 (Table 7).

**Q(12):** Please refer to pages 27-28 lines 424-429, page 28 lines 436-443, pages 28-29 lines 445-448 (Table 5) and page 34 lines 520-524 (Table 7).

**Q(13):** Please refer to pages 28-29 lines 445-448 (Table 5) and page 34 lines 520-524 (Table 7).

**Q(14):** Please refer to page 27 line 410.

**Q(15):** Please refer to page 27 lines 418-423.

**Q(16):** Please refer to page 29 lines 453-455, page 32 lines 483-492 (Figs.7 and 8) and pages 41-42 lines 644-674 (**Section 5.3**).

**Q(17):** Please refer to pages 30-31 lines 477-479 (Table 6).

**Q(18):** Please refer to pages 24-26 lines 359-388 (**Section 3.1.3**) and pages 35-36 lines 525-560 (**Section 4.3**).

Q(19): Please refer to page 11 lines 144-147 (Fig.1), pages 14-15 lines 200-212.

**Q(20):** Please refer to page 39 line 619.

**Q(21):** Please refer to pages 39-40 lines 611-643 (Section 5.2).

**Q(22):** Please refer to page 21 line 297, page 21 lines 305-308, page 22 lines 317-320, page 23 line 338, page 23 line 351, page 25 line 374 and page 40 line 639.

**Q(23):** Please refer to pages 35-36 lines 525-560 (**Sections 4.3**) and pages 37-39 lines 562-610 (**Section 5.1**).

**Reviewers#2:**

Main comments: Please refer to page 4 lines 49-58, pages 7-10 lines 124-129 (Table 1) and pages 19-20 lines 270-279.

**Q(1):** Please refer to pages 6-7 lines 109-123 and pages 41-42 lines 644-674 (**Section 5.3**).

**Q(2):** Please refer to page 11 lines 144-147 (Fig.1), page 15 lines 210-212 (Fig.2), page 17 lines 243-246 (Fig.3), page 18 lines 247-249 (Fig.4), page 20 lines 276-279 (Fig.5), page 31 lines 480-482 (Fig.6), page 32 lines 483-487 (Fig.7), page 32 lines 488-492 (Fig.8) and page 33 lines 493-495 (Fig.9).

**Q(3):** Please refer to page 4 lines 61-63 and page 5 line 67, page 5 lines 74-75 and page 11 line 143.

Q(4): Please refer to page 5 lines 74-75.

**Q(5):** Please refer to page 4 lines 62-63 and page 5 line 67.

**Q(6):** Please refer to page 9 line 126 and page 53 lines 901-903.

**Q(7):** Please refer to pages 4-10 lines 45-129 (Introduction).

**Q(8):** Please refer to pages 5-6 lines 77-84 and pages 22-23 lines 325-336.

**Q(9):** Please refer to page 11 lines 144-147 (Fig.1) and pages 12-13 lines 151-173 (**Section 2.2**).

**Q(10):** Please refer to page 11 lines 144-147 (Fig.1).

**Q(11):** Please refer to page 11 lines 144-147 (Fig.1) and pages 15-16 lines 214-242.

**Q(12):** Please refer to page 4 lines 52-58 and page 19 lines 260-274.

**Q(13):** Please refer to pages 13-14 lines 187-195, pages 15-16 lines 214-242 and page 18 lines 251-254 (Table 3).

**Q(14):** Please refer to pages 15-16 lines 214-230 and page 17 lines 243-246 (Fig.3).

**Q(15):** Please refer to page 16 lines 231-242 and page 18 lines 247-249 (Fig.4).

**Q(16):** Please refer to pages 15-16 lines 221-230 and page 17 lines 243-246 (Fig.3).

**Q(17):** Please refer to page 11 lines 144-147 (Fig.1), pages 13-14 lines 181-195 and page 18 lines 251-254 (Table 3).

Q(18): Please refer to page 5 lines 70-76.

**Q(19):** Please refer to pages 19-20 lines 256-279 and page 22 lines 323-328.

**Q(20):** Please refer to page 32 lines 483-487 (Fig. 7).

**Q(21):** Please refer to pages 25-26 lines 377-388 and pages 35-36 lines 525-560 (**Section 4.3**).

**Q(22):** Please refer to pages 29 lines 453-455, pages 33-34 lines 496-524 (**Section 4.2**), pages 37-39 lines 562-610 (**Section 5.1**) and pages 41-42 lines 644-974 (**Section 5.3**).

**Q(23):** Please refer to pages 4-10 lines 45-129 (**Introduction**) and pages 41-42 lines 644-674 (**Section 5.3**).

**Responses to the Reviewers:**

**Reviewer #1: This manuscript needs to below corrections:**

The paper presented by Li et al. deals with the internal organisation of hydrological systems in terms of the number of reservoirs involved, interactions between these reservoirs and their relative contributions. It is a classical conceptual approach comparable to that of global hydrological models but enriched here by the contribution of tracer data. The article is well written overall, well structured and the illustrations are of good quality (except for figures 6 and 7 which are difficult to read because of the chosen scales). The objectives are clearly stated and the methods used are appropriate and sound. This approach is not, however, original and is a contribution to the series of studies that have been carried out for several years on the contribution of isotopic data to improving the structure of hydrological models (see Uhlenbrook S, Leibundgut C. 1999 for one of the first studies in recent advances). The list of references appears well balanced at first glance with about one third of the references cited being less than 5 years old and half being less than 10 years old. There is little very recent literature on the understanding and modelling of karst systems or on coupled flow-isotope modelling involving questions on mixing processes, residence time distribution or the relationship between velocity and celerity. On the other hand, one third of the articles cited that are less than 5 years old already concern the basin studied (+ 2 other older articles), one of which mentions a coupled hydrology-isotope model. The topic is therefore promising, but we must ask ourselves how this new study improves our knowledge of the system and whether we have made any progress in terms of conceptualization. Apart in the introduction, this question is never addressed and as it stands it does not seem that a totally convincing conceptual scheme has been proposed. In particular, there is too great a disconnection with the field. Beyond the relative adequacy with the flow and isotope data, how does the structure of the model match the morphology of the catchment, underground and on the surface? (A broader discourse is also missing. The authors partly answer the initial question (line 111) but this only concerns the micro site studied. Can the proposed structure be generalised to larger areas (in comparable cockpit karst contexts)?

**Reply:**

We thank the reviewer for their valuable comments and suggestions. In the revision we have cited other more recent publications associated with the coupled flow-isotope modelling of karst system and associated contents.

Cockpit karst landscapes are common in the tropics and sub-tropics area. The cockpit karst covers an area of about 140,000-160,000 km2 in China. Such karst morphology also exists in Southeast Asia, Central America and the Caribbean. Our selected catchment of Chenqi is a karst experimental catchment focused on investigations of hydrological, ecological and geological (carbonate dissolution) changes under climate change and human activities. So there are detailed observational data and field investigations in this catchment. The relevant publications cited in this study are necessary to provide the background context of our new model development and

analysis. Although Chenqi catchment is small, the geomorphologic characteristics can represent a broad region of headwater catchments in cockpit karst landscapes.

In the polje/tower karst areas, the depression is more interconnected with isolated towers scattered throughout the terrain (Lyew et al., 2007). Geological surveys and observations show the hillslope unit lacks surface flow, and the depression unit has surface and underground drainage networks in such karst areas, including our study catchment. Understanding of interconnections of flow systems are vital for developing conceptual hydrological models for cockpit karst landscapes. Our new model presented in the paper is based on the coupled hydrology-isotope model developed by Zhang et al. (2019), a co-author of this manuscript. In this earlier model, the cockpit karst catchment was divided into two morphological units (hillslope and depression) and three water storage compartments (reservoirs) (hillslope reservoir, fast flow and slow flow reservoirs in depression). We substantially improved the model structure with a binary flow system (fast flow and slow flow) in the hillslope unit, and the functioning of a binary moisture storage system of unsaturated zone (see Fig. 4 in the original manuscript). Moreover, we optimized the model structure with a varying number of passive storages at different positions of the flow system (e.g. fast/slow flow reservoirs combined with different hillslope/depression units) based on a multi-objective optimization algorithm for best matching detailed observational data of hydrological processes and isotope concentration in the Chenqi catchment.

We agree there are various connections between hillslope and depression fast/slow flow reservoirs, and the model structure can be further improved in terms of the geomorphological surveys of the catchment. So, we set another reasonable connection between hillslope and depression fast flow and slow flow systems, and re-calibrated and validated the model (see descriptions below). We referenced the previous investigated results and will show more detailed geomorphological data in the revision (e.g. electrical resistivity tomography (ERT) image in Fig. S1) to show how data has informed the evolution of this new model.

**Figure S1**. ERT image in the study depression. They interpret the ERT results as (a) an upper layer consisting of moist soils or extensively fractured rock (marked in blue); (b) carbonate rock with a high secondary porosity (and hence permeability; marked in light blue/yellow); (c) an underlying carbonate rock with low secondary porosity and hence relatively low permeability (marked in red) (Chen et al., 2018).

**Additional reference:**

Lyew-Ayee, P., Viles, H, A., Tucker, G, E.: The use of GIS-based digital morphometric techniques in the study of cockpit karst, Earth Surf. Process. Landforms., 32, 165-179, https://doi.org/10.1002/esp. 1399, 2007.

We have made the necessary changes as advised. Please refer to the page 2 lines 35-41, page 4 lines 51-58, page 5 lines 84-97, pages 37-39 lines 562-610 (Section 5.1) and pages 41-42 lines 644-674 (Section 5.3).

**Q(1) Lines 60-65: the list of references could be extended by some more recent articles (**

---

## Author Response (AR2)

**Title: Effects of passive storage conceptualization on modelling hydrological function and isotope dynamics in the flow system of cockpit karst landscape**
**Author(s): Guangxuan Li, Xi Chen, Zhicai Zhang, Lichun Wang, and Chris Soulsby**
**MS No.: hess-2021-492**
**MS type: Research article**
* * *
**Responses to the Editor:**

Thank you for your letter and the reviewers' comments concerning our manuscript entitled "Effects of passive storage conceptualization on modelling hydrological function and isotope dynamics in the flow system of cockpit karst landscape". We have revised the manuscript according to the reviewers' comments. The revised sentences and sections in the revised manuscript are highlighted in blue color.

Thank you for your editorial work.

Sincerely,

Xi Chen
On behalf of all co-authors

Xi Chen, Professor of Hydrology
Institute of Surface-Earth System Science,
Tianjin University, Tianjin 300072, China
E-mail: xi_chen@tju.edu.cn
* * *
**Responses to the Reviewer #1:**

**I thank the authors for their comprehensive responses and their incorporation into an expanded version of the paper. The elements provided fill the main gaps in the description of the physical context and the justification of the model structure.**

**Q(1) For the description of the physical environment, the improvements made by the authors allow a much better appreciation of the hydrogeological context. In particular, the geological features of the depression are given and the flow conditions in this area are more clearly described. Nevertheless, referring to the work of Chen et al (2018), could you be more specific on the issue of high or low permeability zones in the respective areas of W1 and W4? You mention an aquifer confinement but I rather understood that the permeability reduction was due to a different nature of fracturation (less open fractures? less connected? partially filled?). Please add details in line 189.**

**Reply:**

   The sentences were revised as "W1 is located in a local confined aquifer, consisting of extensively fractured carbonate rock surrounded by rock with low secondary porosity. W4 is located in unconfined aquifer with the vertical permeability reduction from large rock fractures and high secondary porosity to low secondary porosity (Chen et al., 2018)".

Please refer to pages 13-14 lines 188-192.

**Q(2)  I am still not totally convinced by the proposed adjustment lines in figure 3. For example, the correlation relative to point W1 may be statistically significant but it is meaningless when looking at the structure of the scatter plot (a large cluster and some points that pull the relationship). I don't understand why the measurement points in the groundwater and surface water (all points together) do not lie on the local meteoric water line (which by the way is very similar to the global meteoric water line). Even though there is a large scatter, all the points line up correctly on a meteoric water line with a slight deuterium excess. In the context of the study, all waters should be aligned with the LMWL unless there is an evaporation effect or re-condensation of vapour from evaporated local surface water in a low humidity context. I do not think that this second effect is dominant here but if it is the case it could explain the higher deuterium excess observed.**

**Reply:**

   In terms of the plot of $\delta^{18}O$-$\delta D$ for W1, W4, and LMWL (Fig. S1), all waters at W4 and W1 come from rainfall recharge since the line of $\delta^{18}O$-$\delta D$ for W4 is closely aligned with the LMWL and the scattered points of $\delta^{18}O$-$\delta D$ for W1 are concentrated around the LMWL. The agreement of the line of $\delta^{18}O$-$\delta D$ for W4 with the LMWL corresponds to more rainfall recharge into the unconfined aquifer at W4 and fast groundwater flow response to rainfall (refer to Fig. 4 by Chen et al.(2018)). By contrast, the scattered points of $\delta^{18}O$-$\delta D$ for W1 correspond to the confined aquifer at W1 and slow groundwater flow response to rainfall. We revised the associated descriptions.

   We corrected our previous reply about re-condensation. The LMWL does not show re-condensation of vapour from local evapotranspiration (see the line of $\delta^{18}O$-$\delta D$ for precipitation in Fig. 3 in the revised manuscript).

[Figure]

**Figure S1.** Plot of $\delta^{18}O$-$\delta D$ for W1 and W4

Please refer to pages 15-16 lines 222-233.

**Q(3) I still think that there is no really interpretable difference between W1 and W4. Of course, the average difference between the 2 points is slightly above the uncertainty but the number of measurement points is very different. I think the important thing is that the dispersion of the points is greater upstream (W4, point cloud more elongated on the relation in figure 3) than downstream (W1). This probably reflects the effects of a progressive damping of the signal downstream from the mixing of the different water fractions. Nevertheless, this descriptive part is not crucial and does not call into question the core of the work. One possibility would be to drastically reduce this paragraph by simply mentioning the average differences observed (especially between catchment outlet and hillslope spring) and by pointing out the few alterations due to evaporation in surface waters.**

**Reply:**

We have deleted the dispersion of the isotope values (CV in Table 3) since the number of measurement points is very different at sites and the range of isotope values can reflect the variability.

We revised the descriptions in this paragraph. We believe that the use of isotopic signatures to indicate the hydrological functions of flow damping, water evaporation and connection from hillslope spring to catchment outlet has been sufficiently described in the revised manuscript.

Please refer to pages 15-16 lines 222-233 and page 18 lines 254-256 (Table 3).

**Q(4)  The structure of the model is related to the geomorphological context. Once the conceptual model is established, it is easier to understand the choices for the interactions between UZ and SZ and between hillslope and depression. It is interesting to have introduced a parameter allowing a proportion of the water from the hillslope slow reservoir to join the depression slow reservoir. From the new optimization procedure, the optimized ratio rhd was 0,39 for model f. For what I understand, this mean that 61% of the slow flow from the hillslope reaches the depression conduits. If so, the overall concept is therefore significantly different. Is the increase of the fast flows proportion in the depression underground channel still consistent with the observations of the flow dynamics at the underground channel outlet? (not only the proportions of total flow)**

Reply:

The hillslope flow is composed of the most fast flow component (over 80%) and thus the slow flow amount is small (less than 20%). So even though 61% of the slow flow on the hillslope enters the conduit of the depression, the hillslope slow flow only accounts for 13% of the total flow allocated from the hillslope to the depression as shown in Table 8 (the revised manuscript). So, it is still consistent with the observations of the flow dynamics at the underground channel outlet that fast flow is the dominant component in the catchment.

**Q(5)  technical points :**
**1)- Line 178 and line 181 : only one spring I suppose? If so, please leave "spring" in the singular**
**2)- Line 205 : hillslope flow : do you mean hillslope spring discharge ?**
**3)- Line 372 : Would it be appropriate to complete figure 5 by mentioning the parameter Vm? As the surface flow represents almost half of the total discharge, Vm is a major parameter.**

Reply:

We have modified those points according to the comments of the reviewer.

Please refer to page 13 lines 178-181, page 14 lines 207-208 and page 20 lines 278-281 (Fig.5).
* * *
**Responses to Editor Polina Shvedko:**

**Notification to the authors:**
**For your next revision: Please add the country to the second affiliation. Please make sure that all text including citations are in black font. Please ensure that the colour schemes used in your maps and charts allow readers with colour vision deficiencies to correctly interpret your findings. Please check your figures using the Coblis – Color Blindness Simulator (https://www.color-blindness.com/coblis-color-blindness-simulator/) and revise the colour schemes accordingly.**

**Reply:**

We have revised our manuscript according to your requirements.

1) We have added the country to the second affiliation. Please refer to page 1 line 12.
2) We have revised our manuscript to make sure that all text including citations are in black font. Please refer to the revised manuscript and the authors' tracked changes.
3) After checking all the figures with the Coblis - Color Blindness Simulator, we found that there was a problem with the color schemes in Fig.4, so we redraw it. Please refer to page 18 lines 250-252 (Fig.4).

---

## Author Response (AR3)

**Title: Effects of passive storage conceptualization on modelling hydrological function and isotope dynamics in the flow system of cockpit karst landscape**
**Author(s): Guangxuan Li, Xi Chen, Zhicai Zhang, Lichun Wang, and Chris Soulsby**
**MS No.: hess-2021-492**
**MS type: Research article**
* * *
**Responses to the Editor Thom Bogaard:**

Thank you for your letter and comments concerning our manuscript entitled "Effects of passive storage conceptualization on modelling hydrological function and isotope dynamics in the flow system of cockpit karst landscape". We have revised the manuscript according to your comments.

Thank you for your editorial work.

Sincerely,

Xi Chen
On behalf of all co-authors

Xi Chen, Professor of Hydrology
Institute of Surface-Earth System Science,
Tianjin University, Tianjin 300072, China
E-mail: xi_chen@tju.edu.cn
* * *
**Q: Dear authors, thanks for the revision. Thanks for the corrections. I went through the paper and think it is ready for publication except for 1 technical correction. You are using quite a lot of parameters/variables that you abbreviated with two or more letters (PE, wm, ks, WMM, EX, ke, WU, EXM, etc). This we try to avoid as they are confusing (and also incorrect although done often in our literature): wm is w multiplied with m....etcetera). Also I noticed that sometimes it is not clear if it is ks or k_subscript_s). So please replace them with 1 letter symbols and some subscripts. Also do a proper typesetting for the subscripts to prevent misinterpretations). This also holds for some formula's typed in sentences. Please check. Abbreviation like DS, HS, HF, DF etc can remain, they are abbreviation for words, not part of an equation. (Idem KGE)**

**Reply:**

We thank the editor for his valuable comments and suggestions.

We have modified the symbols of all parameters and variables that may cause misunderstanding, as listed in Table S1:

**Table S1.** Summary of changes in the symbols of parameters and variables

| Parameters/Variables | | Meaning |
|---|---|---|
| Original symbol | New symbol | |
| $wm$ | $W_m$ | mean storage capacity |
| $wm$' | $W_{m'}$ | areal mean tension water storage at $f$ |
| $WMM$ | $W_{mm}$ | maximum value of $W_{m'}$ |
| PE | Delete | net precipitation |
| $Ep$ | $E_p$ | potential evapotranspiration |
| $WU$ | $W_U$ | moisture storage consisting of active storage $W$ or mobile water (Sprenger et al., 2017; Sprenger et al., 2018) and passive storage $W_P$ |
| $EX$ | $E_X$ | flux between fast flow and slow flow reservoirs |
| $EXM$ | $E_{XM}$ | exchange mass between the slow flow and fast flow reservoirs |
| $EGM$ | $E_{GM}$ | mixing of the solute between the active and passive storages for the slow or fast flow reservoirs |
| $kc$ | $k_c$ | coefficient for evapotranspiration |
| $ks$ | $k_s$ | ratio of water yield into slow flow reservoir |
| $ke$ | $k_e$ | exchange coefficient between slow and fast flow reservoirs |
| $\eta s$ | $\eta_S$ | outflow coefficient of slow flow reservoir |
| $\eta f$ | $\eta_F$ | outflow coefficient of fast flow reservoir |
| $ls$ | $l_s$ | coefficient of evaporation fractionation |

In addition, we use "$\times$" to represent the "multiply" sign to avoid misunderstanding.

Please refer to pages 19-29 lines 257-449 (**Section 3**), pages 33-34 lines 498-525 (**Section 4.2**) and page 40 lines 633-640.